# World2Minecraft: Occupancy-Driven Simulated Scenes Construction

**Lechao Zhang[1], Haoran Xu[1], Jingyu Gong[1], Xuhong Wang[2], Yuan Xie[1,3], Xin Tan[1,2,*]**

[1]School of Computer Science and Technology, East China Normal University
[2]Shanghai Artificial Intelligence Laboratory
[3]Shanghai Innovation Institute
{51275901049, 10235102427}@stu.ecnu.edu.cn
wangxuhong@pjlab.org.cn, {jygong, yxie, xtan}@cs.ecnu.edu.cn

## Abstract

Embodied intelligence requires high-fidelity simulation environments to support perception and decision-making, yet existing platforms often suffer from data contamination and limited flexibility. To mitigate this, we propose *World2Minecraft* to convert real-world scenes into structured Minecraft environments based on 3D semantic occupancy prediction. In the reconstructed scenes, we can effortlessly perform downstream tasks such as Vision-Language Navigation(VLN). However, we observe that reconstruction quality heavily depends on accurate occupancy prediction, which remains limited by data scarcity and poor generalization in existing models. We introduce a low-cost, automated, and scalable *data acquisition pipeline* for creating customized occupancy datasets, and demonstrate its effectiveness through *MinecraftOcc*, a large-scale dataset featuring 100,165 images from 156 richly detailed indoor scenes. Extensive experiments show that our dataset provides a critical complement to existing datasets and poses a significant challenge to current SOTA methods. These findings contribute to improving occupancy prediction and highlight the value of *World2Minecraft* in providing a customizable and editable platform for personalized embodied AI research. **Project page:** https://world2minecraft.github.io/.

## 1 Introduction

Embodied intelligence aims to develop intelligent agents that can perceive, understand, and interact within complex environments. Progress in this field depends critically on the availability of high-fidelity, diverse simulation environments supported by robust datasets. Platforms like Habitat (Savva et al., 2019) is limited by their reliance on real-world scans, which not only yield scenes with visual and geometric artifacts but uneditable, limiting their utility for agents that need to modify their environment. Minecraft is widely used for reinforcement learning (Cai et al., 2023; Li et al., 2025; Cai et al., 2024b; Zheng et al., 2025a; Cai et al., 2025b) for its customizable environments. However, its blocky graphics create a stark reality gap despite recent domain-invariant prompting efforts (Zhao et al., 2024). These limitations highlight the need for new simulation platforms that are both highly flexible and editable, and also capable of maintaining visual realism.

Real-to-sim transfer presents an effective approach for this goal. However, current techniques like NeRF (Mildenhall et al., 2021) and 3D Gaussian Splatting (Kerbl et al., 2023; Ji et al., 2025; Tian et al., 2025) often yield photorealistic views but uneditable representations that lack physical properties. Similarly, CAD-based methods (Avetisyan et al., 2019; Gümeli et al., 2022; Tyszkiewicz et al., 2022; Murali et al., 2017) yield clean scenes but require precise instance segmentation and can not be directly used for downstream tasks. To reconcile realism with interactability, we utilize 3D semantic occupancy (Cao & De Charette, 2022). Unlike implicit fields, its discrete voxel structure naturally aligns with Minecraft blocks. This compatibility enables the direct translation of real-world scenes into editable, physically grounded environments, bypassing complex mesh-to-block conversions.

Inspired by this, we propose *World2Minecraft*, a framework that reconstructs real-world scenes as high-quality Minecraft environments by leveraging 3D semantic occupancy prediction as shown in Figure 1. In contrast to existing methods, our approach is cost-effective, yields readily editable scenes, and is directly applicable to downstream tasks such as Vision-Language Navigation

---

*Corresponding author.

(VLN) (Anderson et al., 2018; Gu et al., 2025). The framework operates by first predicting single-frame 3D semantic occupancy, then integrating multi-frame observations via camera parameters (Wu et al., 2024) to build a unified semantic occupancy field of the complete scene. The resulting representation can be refined via a developed visual tool(as shown in Appendix J) before generating construction instructions for Minecraft. Executing these instructions faithfully reproduces the high-fidelity scene in Minecraft.

After reconstructing real-world scenes in Minecraft, we conducted extensive experiments on downstream VLN tasks. To this end, we constructed ***MinecraftVLN***, a dataset composed of 1,059 samples from our reconstructed scenes, augmented with 2,483 additional samples from community-created large-scale scenes to increase scale and diversity. We defined two subtasks—**Next-View** and **Next-Action**—and fine-tuned Qwen2.5-VL-3B (Bai et al., 2025) and Qwen2.5-VL-7B (Wang et al., 2024) on each, achieving notable performance gains. Real-time navigation was successfully demonstrated by employing Gemini-2.5-Pro (Comanici et al., 2025) as the controller for an agent in the reconstructed Minecraft environments. However, the reconstruction quality remained sub-optimal for practical use. We identified that accurate semantic occupancy prediction is critical to reconstruction fidelity and scalability, yet it faces two major limitations: (1) heavy reliance on large-scale, expensively annotated data (?), and (2) dataset constraints such as limited diversity, poor coverage, and sensor noise (Liu et al., 2023), which hinder model generalization in complex real-world scenarios.

To advance the generalization of scene occupancy prediction, we introduce a novel, low-cost, and automated pipeline for generating customized semantic occupancy datasets, which significantly reduces the traditional reliance on expensive manual annotation or limited real-world scans. We demonstrate its effectiveness through ***MinecraftOcc***, a large-scale dataset produced by this pipeline, featuring 100,165 high-resolution images captured from continuous roomtour across 156 richly detailed indoor scenes constructed in Minecraft. By leveraging mods for physically-based rendering and precise layout control, we automatically generate visually realistic environments with complex structures, diverse objects, and dynamic lighting, effectively narrowing the sim-to-real gap. Experiments show that current occupancy models perform poorly on ***MinecraftOcc***, revealing clear generalization limits. Moreover, when used as auxiliary training data, it enhances performance on real-world benchmarks like NYUv2(Silberman et al., 2012), confirming its dual role as a challenging benchmark and an effective data augmentation resource for improving model robustness. In summary, the main contributions of this paper are as follows:

- We introduce ***World2Minecraft***, a pipeline for real-world scene reconstruction in Minecraft via semantic occupancy prediction.

- We conduct VLN task within the reconstructed scenes, during which we construct the ***MinecraftVLN*** dataset to validate the practical utility of our approach.

- We propose an automated pipeline for semantic occupancy data generation, and present the large-scale ***MinecraftOcc*** benchmark, which exposes the generalization limits of existing methods and serves as effective training data for enhancing robustness.

## 2 RELATED WORK

### 2.1 DATA-DRIVEN 3D SCENE GENERATION

In embodied intelligence research, real-to-sim transfer—converting real-world scenes into simulated environments—remains a critical yet challenging task. Generative approaches, such as 3D-GPT (Sun et al., 2025a), SceneCraft (Yang et al., 2024) and Styleshot (Gao et al., 2025), excel in creating novel content from abstract inputs but not designed to faithfully reconstruct specific, existing real-world locations.While WonderWorld (Yu et al., 2025) generates 3D worlds from a single image but lack the semantic decomposability and editability. Similarly, CAD-based methods (Avetisyan et al., 2019; Gümeli et al., 2022; Tyszkiewicz et al., 2022; Murali et al., 2017) yield clean and lightweight scene representations. However, they rely heavily on precise instance segmentation and accurate scale alignment between the retrieved CAD models and the real-world scene, which hinders their direct application in downstream tasks. Recent methods like LiteReality (Huang et al., 2025) simplify real-to-virtual conversion, yet remain limited in object and scene diversity, and cannot be directly used for downstream tasks. We propose a low-cost and easily editable method for real-to-sim conversion by leveraging occupancy prediction, enabling the direct application of reconstructed environments to downstream tasks like VLN in Minecraft.

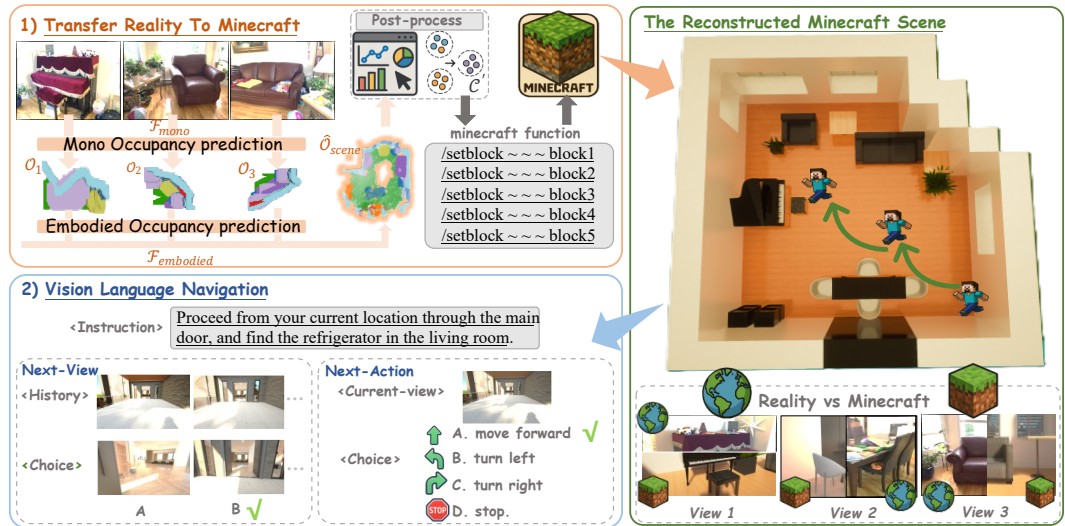

Figure 1: **Framework of World2Minecraft**, which illustrates the process of reconstructing real-world scenes into Minecraft environments and conducting navigation within these scenes. 1) For the transfer reality to Minecraft, RGB images are input into the occupancy prediction model, which is then postprocessed to generate instructions for reconstruction in Minecraft. 2) VLN tasks involving Next-View and Next-Action are performed within the reconstructed scenes.

## 2.2 INDOOR 3D OCCUPANCY PREDICTION DATASET

3D semantic understanding is a fundamental task in computer vision, with extensive research focused on point cloud segmentation (Gong et al., 2021a;b; Sun et al., 2024) and 3D occupancy prediction. However, the development of accurate models is hampered by the scarcity of large-scale, high-quality annotated data. Existing datasets, such as NYUv2 (Silberman et al., 2012), OccScan-Net (Yu et al., 2024), and EmbodiedOcc-ScanNet (Wu et al., 2024), are typically derived from real-world scans. They consequently suffer from limitations including sensor noise, sparse annotations, and constrained object diversity, while being costly and time-consuming to produce. These challenges underscore the need for a more efficient data creation paradigm. In response, we propose an automated and labor-efficient pipeline for synthesizes high-fidelity voxel occupancy at a fraction of the cost, enabling scalable and diverse data generation for robust model training.

## 2.3 EMBODIED INTELLIGENCE RESEARCH IN MINECRAFT

Minecraft has been widely used for embodied intelligence and reinforcement learning research (Cai et al., 2025a; Zheng et al., 2025a; Cai et al., 2024b; Wang et al., 2023), with many works built upon MineStudio (Cai et al., 2024a), a streamlined open-source framework that unifies simulation and data management. To this end, ROCKET-1 (Cai et al., 2025b) leverages visual-temporal context prompting to master open-world interactions, JARVIS-VLA (Li et al., 2025) post-trains large-scale vision-language models to perform diverse in-game tasks with keyboards and mouse, and GROOT (Cai et al., 2023) learns to follow instructions by watching gameplay videos. Despite these advances, all these approaches operate in Minecraft's native blocky visuals, which exhibit a substantial reality gap compared to real-world scenes. To address this limitation, we integrate high-fidelity community mods, significantly narrowing the visual and structural gap and providing a more effective simulation environment for embodied intelligence.

## 3 METHOD

### 3.1 PRELIMINARIES

Minecraft serves as a valuable platform for embodied AI research due to its voxel-based, spatially discretized world and consistent physical mechanics. However, the vanilla game presents limitations for perception-related tasks, including a significant visual domain gap from reality, limited semantic diversity of blocks, and simplistic indoor structures. To address these issues, we developed a customized environment using different mods, including *WorldEdit*, *Screen with Coordinates* and *TMEO*. A detailed introduction to the standard environment and our modifications is provided in Appendix B.

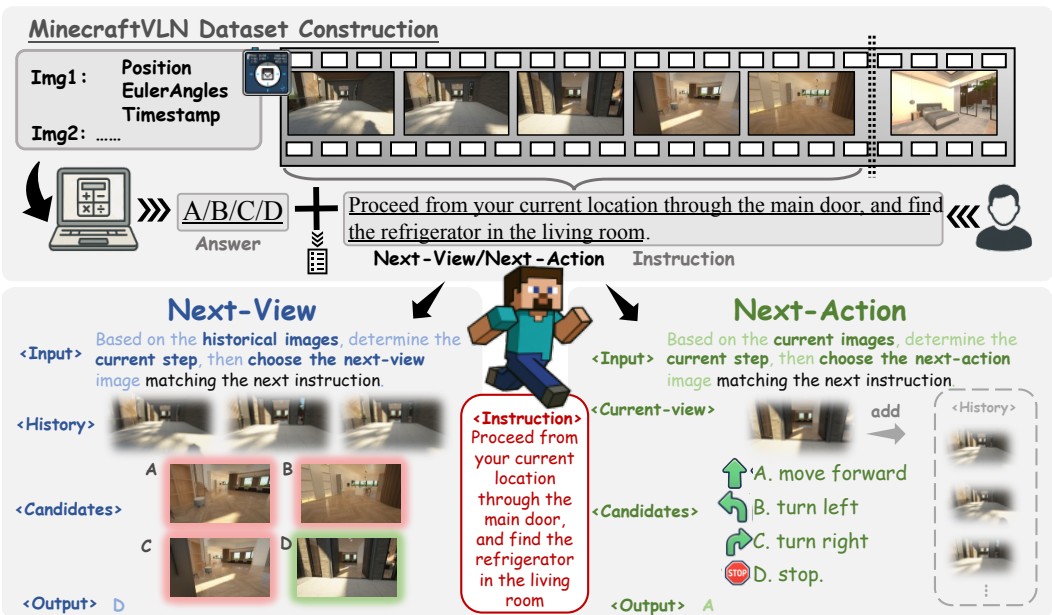

Figure 2: **Dataset Construction Pipeline for MinecraftVLN**. We segment roomtour sequences into valid trajectories, then generate instruction-following Question-Answer pairs using the collected coordinates and orientations to construct Next-View and Next-Action dataset.

## 3.2 WORLD2MINECRAFT: TRANSFER REALITY TO MINECRAFT

We propose ***World2Minecraft***, a framework that converts real-world scenes into Minecraft via 3D semantic occupancy prediction (Fig. 1(1)). As detailed in Algorithm 1 (Appendix C), our method establishes an end-to-end pipeline from multi-view perception to executable Minecraft commands. The core of our method addresses 3D semantic occupancy prediction from a sequence of first-person images $\mathcal{I} = \{I_1, I_2, \ldots, I_N\}$ along with their corresponding camera intrinsic parameters $\mathcal{K}$. First, we employ a monocular predictor $\mathcal{F}_{\text{mono}}$ that generates per-view semantic occupancy grids $\mathcal{O}^i_{\text{mono}}$ from individual RGB images, where each voxel is assigned a semantic label from C total classes:

$$\mathcal{O}^i_{\text{mono}} = \mathcal{F}_{\text{mono}}(I_i, \mathcal{K}) \in \{0, 1, \ldots, C-1\}^{X \times Y \times Z}. \tag{1}$$

When the per-image occupancy predictions are obtained, we leverage camera extrinsic parameters $\mathcal{E}$ to merge them into a unified 3D semantic representation $\hat{\mathcal{O}}_{\text{scene}}$:

$$\hat{\mathcal{O}}_{\text{scene}} = \mathcal{F}_{\text{embodied}}\left(\{\mathcal{O}^i_{\text{mono}}\}_{i=1}^N, \mathcal{K}, \mathcal{E}\right) \in \{0, 1, \ldots, C-1\}^{X \times Y \times Z} \tag{2}$$

To identify potential object centers, we first convert the multi-class semantic grid $\hat{\mathcal{O}}_{\text{scene}}$ into a binary occupancy grid $\hat{\mathcal{O}}_{\text{binary}}$, where voxels corresponding to any object class are marked as 1 and empty voxels as 0. We then compute a local occupancy density map $\mathcal{D}$ on this binary grid by applying a 3D convolution with a uniform kernel $\mathbf{K} \in \mathbb{R}^{k \times k \times k}$. Potential centers $\mathcal{C}$ are identified by applying a density threshold $\tau$:

$$\mathcal{C} = \left\{\mathbf{v} \,\middle|\, \mathcal{D}(\mathbf{v}) \geq \tau, \mathcal{D} = \mathbf{K} * \hat{\mathcal{O}}_{\text{binary}}, \mathbf{v} \in \hat{\mathcal{O}}_{\text{scene}}\right\} \tag{3}$$

where $*$ denotes the 3D convolution, and $\mathbf{v} = (x, y, z)$ represents voxel coordinates. These initial center points $\mathcal{C} = \{\mathbf{c}_j\}_{j=1}^M$ are often redundant. To obtain a accuarte and representative set of object locations, we cluster these points using the DBSCAN (Ester et al., 1996) algorithm. Clustering is performed independently for the center points within each semantic class. This ensures that points with different semantic labels are not grouped together, preserving the categorical integrity of objects. It groups points based on a distance threshold $\eta$, using the L2 norm as the metric. Each resulting cluster is then represented by its centroid, forming a refined set of centers $\mathcal{C}'$:

$$\mathcal{C}' = \left\{\frac{1}{|\mathcal{G}_\mu|} \sum \mathbf{v} \in \mathcal{G}_\mu \,\middle|\, \mathcal{G}(\mu) = \{\mathbf{v} \in \mathcal{C} \mid |\mathbf{v} - \mu|_2 \leq \eta\}, \mu \in \mathcal{C}\right\} \tag{4}$$

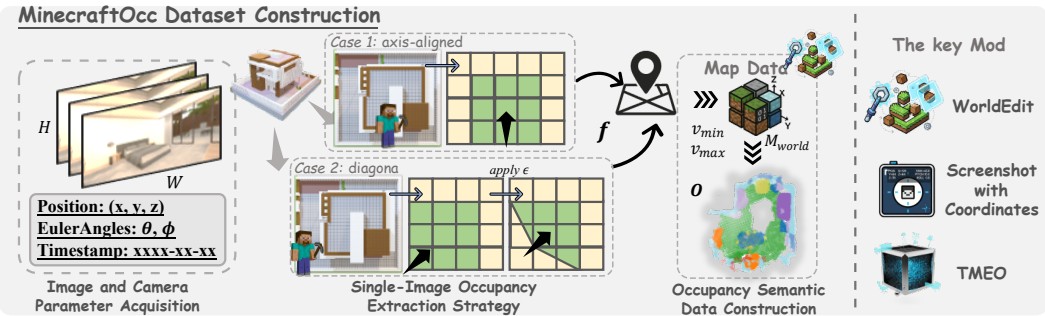

Figure 3: **Dataset Construction Pipeline for MinecraftOcc**. We record coordinate data during roomtour, divide the viewpoint into two yaw-based cases to define view regions(the yellow indicates invisible areas; green indicates visible areas), and extract semantic occupancy from map data.

This process yields a refined set of centers $\mathcal{C}' = \{\mathbf{c}'_k\}_{k=1}^K$ ($K \leq M$), where each centroid identifies a distinct object instance. Before final rendering, to ensure geometric fidelity, we employ a retrieval-based matching mechanism. Specifically, we align each instance's occupancy grid $\mathcal{O}_k$ with a candidate furniture library $\mathcal{L} = \{\mathbf{T}_j\}_{j=1}^M$. We address orientation ambiguity by iterating through a discrete set of rotation angles $\delta$, selecting the optimal template and rotation that maximize spatial overlap:

$$(j^*, \delta^*) = \underset{j,\delta}{\operatorname{argmax}} \frac{|\mathcal{O}_k \cap \operatorname{Rot}(\mathbf{T}_j, \delta)|}{|\mathcal{O}_k \cup \operatorname{Rot}(\mathbf{T}_j, \delta)|} \tag{5}$$

Once the optimal geometric representations are retrieved, they are translated into Minecraft building commands to render the complete virtual scene.

### 3.3 ENABLING VLN IN MINECRAFT

**Preparations.** To conduct VLN in the reconstructed scene within Minecraft, we reconstructed all real-world indoor scenes from the validation set of EmbodiedOcc-ScanNet dataset (Wu et al., 2024) in Minecraft. Due to the limited accuracy of current prediction models, we selected 15 scenes for manual refinement to ensure high fidelity. However, we observed that the limited scale of the reconstructed scenes resulted in a navigation dataset with relatively short and simple instructions. To address this, we incorporated 5 additional community-created Minecraft scenes, thereby increasing the complexity and diversity of the instruction set (as shown in Figure 6 and Table 8). An agent was subsequently directed to perform room tours within these 20 selected scenes, generating a series of image sequences annotated with positions and orientations.

**MinecraftVLN Dataset Construction.** As shown in Figure 2, the dataset was constructed by processing trajectories from the roomtour image sequence to extract meaningful navigation segments. Human annotators provided detailed textual descriptions for each segment. Using a question-answering template, we generated 3,801 items in total(as shown in Table 9): 1,059 samples (***Base***) from the real-world reconstructed scenes and 2,483 samples (***Extend***) from the community-created scenes. The combined dataset (***Combined***) merges the aforementioned ***Base*** and ***Extend*** sets.

The ***MinecraftVLN*** dataset includes two distinct tasks: 1)**Next-View Prediction**: The agent receives a natural language instruction and a sequence of three historical images. It must first localize its current navigation step based on the instruction and visual context, and then predict the next most probable view. 2)**Next-Action Prediction**: Given the instruction and the current view, the agent identifies its current progress within the instruction and predicts the next action (e.g., move forward, turn left) to comply with the instruction.

**Conduct VLN in Minecraft.** We conducted experiments in two key directions. First, we fine-tuned Qwen2.5-VL (Wang et al., 2024; Bai et al., 2025) on the ***MinecraftVLN***. Second, we deployed Gemini-2.5-Pro (Comanici et al., 2025) for direct embodied navigation control in Minecraft. Detailed experimental setups and results are presented in Sec. 4.3.

Table 1: Comparison between MinecraftOcc, NYUv2, and OccScanNet across key statistics.

| Dataset | Num. of Images | Num. of Scenes | Num. of Classes | Total Semantic Voxels | Avg. Voxels per Scene | Image Resolution |
|---|---|---|---|---|---|---|
| NYUv2 | 1,449 | 464 | 13 | 10,786,528 | ~23.2K | 640 × 480 |
| OccScanNet | 65,119 | 674 | 13 | 201,215,233 | ~298.5K | 640 × 480 |
| MinecraftOcc | 100,165 | 156 (~1,000 rooms) | 1,452 | 733,280,256 | ~4.7M | 1920 × 1129 |

## 3.4 MINECRAFTOCC DATASET CONSTRUCTION

In this section, we will detail our automated semantic occupancy generation pipeline and the MinecraftOcc dataset built upon it.

**Problem Definition.** The 3D occupancy prediction task (as detailed in Sec. 3.2) infers the geometric and semantic structure of a scene from a set of first-person images $\mathcal{I}$, along with their camera intrinsic parameters $\mathcal{K}$ and extrinsic parameters $\mathcal{E}$.

**Image and Camera Parameter Acquisition.** To acquire our dataset, we used an automated mod tool, *Screen with Coordinates*, which simultaneously captures first-person screenshots and records the corresponding camera pose (3D position and orientation). With this paired data of images and poses, we compute the corresponding intrinsic and extrinsic camera matrices for each image. The detailed methodology for deriving these matrices from the virtual camera's Field of View (FOV), position, and orientation is elaborated in Appendix D.

**Single-Image Occupancy Extraction Strategy.** To generate semantic occupancy labels for each image, we define a fixed-size 3D spatial volume $\mathcal{V}$, with a minimum corner $\mathbf{v}_{\min}$ and a maximum corner $\mathbf{v}_{\max}$. To accurately define the 3D spatial region based on the player's viewpoint, we categorize the player's yaw angle $\theta$ into two fundamental cases (in fact, all possible viewpoints can be ultimately categorized into these two cases) based on its orientation relative to the world grid: axis-aligned(as shown *Case 1* in Figure 3), where the viewpoint is parallel to a coordinate axis, and diagonal(as shown *Case 2* in Figure 3), where the viewpoint is directed along a 45-degree angle to the axes. For the axis-aligned case, we set the player's position $P_{\text{player}} = (x_p, y_p, z_p)$ as the center of the $\mathcal{V}$'s back face. For the diagonal case, we set the $P_{\text{player}}$ as the volume's minimum corner $\mathbf{v}_{\min}$. This logic is formalized by the calculation function $f$:

$$(\mathbf{v}_{min}, \mathbf{v}_{max}) = f(P_{player}, \theta, w, h, d) \tag{6}$$

where $(w, h, d)$ are the dimensions of the volume, and each point $v \in \mathcal{V}$ satisfies the boundary constraints $v_{\min} \leq v \leq v_{\max}$.

Due to the discrete nature of Minecraft's space, diagonal views often suffer from significant voxel loss at the periphery. To mitigate this, we designed a viewpoint-aware fallback strategy. This strategy supplements structural information from slightly adjusted neighboring viewpoints based on the original view, significantly enhancing the completeness of the data labels and improving stability and robustness during training. Specifically, we apply a correctional offset, $\boldsymbol{\epsilon}$, to the bounding box corners $\mathbf{v}_{\min}$ and $\mathbf{v}_{\max}$:

$$\begin{aligned} \mathbf{v}'_{\min} &= \mathbf{v}_{\min} + \boldsymbol{\epsilon} \\ \mathbf{v}'_{\max} &= \mathbf{v}_{\max} + \boldsymbol{\epsilon} \end{aligned} \tag{7}$$

This adjustment expands the player's visible range, resulting in a field of view that more accurately reflects their actual perspective, while sacrificing only a minimal portion of the depth-of-field area—a negligible loss in terms of overall visual coverage.

**Occupancy Semantic Data Construction.** To obtain semantic labels, we utilized the *WorldEdit* mod to extract the block types at these coordinates from the map data. This can be viewed as querying a world-map function, $M_{world}$, which maps any coordinate $\mathbf{v}$ to a semantic label $s$ from a set of all possible block types $\mathcal{S} = \{air, stone, wood, ...\}$, which allowed us to construct the final voxel-level semantic occupancy representation, a grid $O$:

$$s_{\mathbf{v}} = M_{world}(\mathbf{v}), \quad \forall \mathbf{v} \in \mathcal{V} \tag{8}$$

$$O = \{s_{\mathbf{v}} \mid \mathbf{v} \in \mathcal{V}\} \tag{9}$$

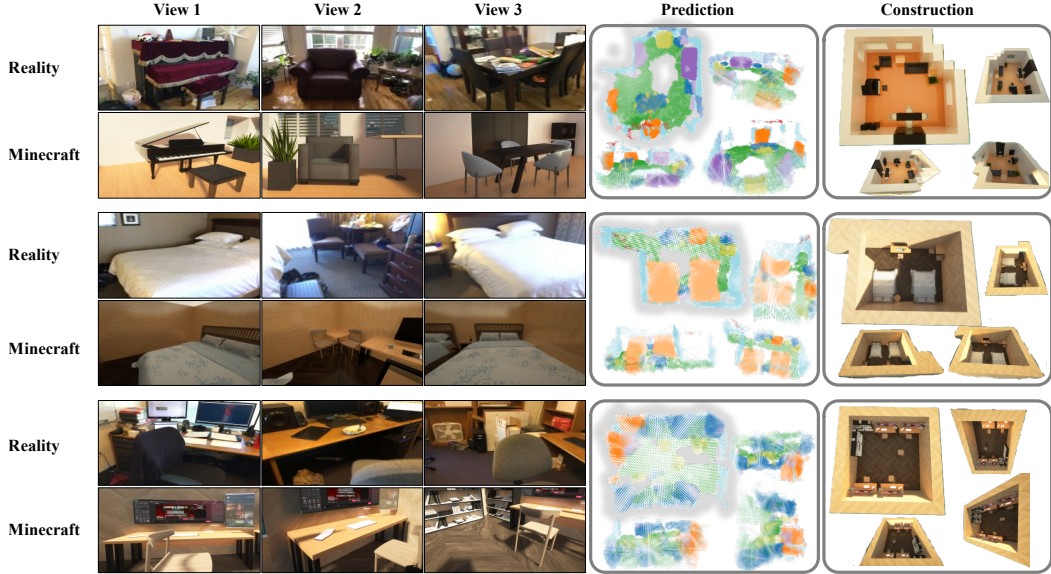

Figure 4: The reconstruction results from reality to Minecraft are presented above. As we can observe that from View 1 to View 3, the Reality row and the Minecraft row demonstrate a high degree of consistency. The Prediction column displays the predicted occupancy views from different perspectives of the same scene, while the corresponding reconstructed scenes in the Construction column align well with them.

## 4 EXPERIMENTS

### 4.1 EXPERIMENTAL SETUP

**Datasets.** Our experimental setup employs multiple datasets to evaluate different aspects of the proposed method and dataset. We employ the EmbodiedOcc-ScanNet dataset (Wu et al., 2024) to train occupancy prediction model, which is the core of the ***World2Minecraft***. For evaluating downstream task performance, we introduce the ***MinecraftVLN*** dataset including Next-View and Next-Action. We also introduce ***MinecraftOcc*** to evaluate the existing methods and employ the NYUv2 dataset for comparative analysis and mixed-training experiments.

**Evaluation Metrics.** We employ standard metrics for each aspect of our evaluation: Accuracy for the VLN task, mean Intersection over Union (mIoU) and Intersection over Union (IoU) for occupancy prediction, and no-reference image quality metrics including Natural Image Quality Evaluator (NIQE), Perception-based Image Quality Evaluator (PIQE), and Laplacian Variance (LV) to assess dataset realism (see Appendix H for details).

### 4.2 THE RESULTS OF WORLD2MINECRAFT

**Implementation Details.** We employed the pre-trained EmbodiedOcc (Wu et al., 2024) model to reconstruct real-world scenes from the full validation set of EmbodiedOcc-ScanNet (Wu et al., 2024) in Minecraft. From these reconstructions, we selected 30 scenes for meticulous manual refinement to enhance their structural completeness and visual quality. Among these, 15 high-quality scenes were chosen to construct the ***MinecraftVLN*** dataset, which also serves as the platform for developing and evaluating embodied navigation agents controlled by large language models. As the initial automated reconstructions were suboptimal due to inherent model limitations, manual refinement effectively restored geometrically consistent layouts, particularly in the accurate placement of modern indoor furniture, as demonstrated in the resulting scenes.

**Analysis.** The reconstruction results from reality to Minecraft are presented in Figure 4. As we can observe that from View 1 to View 3, the Reality row and the Minecraft row demonstrate a high degree of consistency. The Prediction column displays the predicted occupancy views from different perspectives of the same scene, while the corresponding reconstructed scenes in the Con-

Table 2: Across three distinct MinecraftVLN settings, the performance (Accuracy) of Qwen2.5-VL models (3B and 7B) on Next-View and Next-Action tasks under No Training, SFT, and RFT conditions is evaluated.

| Dataset Composition | Task | Qwen2.5-VL-3B | | | Qwen2.5-VL-7B | | |
|---|---|---|---|---|---|---|---|
| | | No Train | SFT | RFT | No Train | SFT | RFT |
| Base | Next-View | 0.2195 | 0.5610 | 0.2927 | 0.3905 | **0.5854** | 0.4390 |
| | Next-Action | 0.1943 | 0.7200 | 0.6343 | 0.3829 | **0.8000** | 0.6343 |
| Extend | Next-View | 0.2261 | **0.7087** | 0.3043 | 0.2913 | 0.6826 | 0.6043 |
| | Next-Action | 0.3657 | 0.5437 | **0.6667** | 0.3786 | 0.6019 | 0.6343 |
| Combined | Next-View | 0.2288 | 0.5609 | 0.3137 | 0.2878 | 0.6642 | **0.6753** |
| | Next-Action | 0.3037 | 0.4835 | **0.6570** | 0.3760 | 0.6281 | 0.6219 |

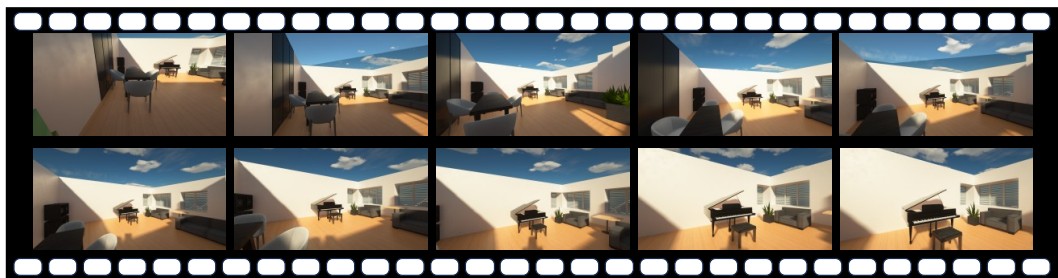

Figure 5: The result of a Gemini-2.5-Pro controlled agent performing VLN in our reconstructed scene. Following the natural language instruction "Go to the piano", the agent successfully navigates to the target step by step.

struction column align well with them, which collectively demonstrates the effectiveness of our ***World2Minecraft*** pipeline.

### 4.3 THE RESULT OF VLN IN MINECRAFT

**Implementation Details.** We conducted VLN in the reconstructed Minecraft environment. We first collected a navigation dataset within the 15 scenes mentioned in Sec. 4.2. However, we observed that the limited scale of the reconstructed scenes resulted in a navigation dataset with relatively short and simple instruction sequences (denoted as ***Base***). To mitigate this, we extended data collection to community-built Minecraft scenes (denoted as ***Extend***), thereby increasing the complexity and diversity of the instruction set (as shown in Figure 6 and Table 8). The combined dataset (denoted as ***Combined***) was then used for evaluation. We evaluated two subtasks, Next-View and Next-Action, across all three dataset settings. For each setting, experiments were conducted under three conditions: No training, Supervised Fine-Tuning (SFT) based on LLama-Factory (Zheng et al., 2024), and Reinforcement Fine-Tuning (RFT) based on EasyR1 (Zheng et al., 2025b), adopting Qwen2.5-VL(3B and 7B) models as the base model. The results are summarized in Table 2.

**Analysis.** The experimental results confirm that, as a baseline, the 7B model outperforms the 3B model in zero-shot settings, and both SFT and RFT lead to significant performance gains. However, the optimal fine-tuning strategy is not uniform. On the more challenging multi-image **Next-View** task, SFT proves more effective for the smaller 3B model, while the performance gap between SFT and RFT narrows considerably for the larger 7B model. Conversely, on the **Next-Action** task, the best method depends on the dataset: SFT excels on the ***Base*** set, whereas RFT shows superior performance on the more diverse ***Extend*** and ***Combined*** sets. These results collectively validate the effectiveness of our dataset and demonstrate the feasibility of conducting complex navigation tasks within the reconstructed Minecraft environments.

**Application in Minecraft.** We deploy Gemini-2.5-Pro to control an agent performing VLN in a ***World2Minecraft*** reconstructed scene. The agent successfully locates a target piano by following natural language instructions "Go to the piano" as shown in Figure 5, demonstrating the practical utility of our environment for embodied AI.

### 4.4 Experimental Results about MinecraftOcc

**Implementation Details.** We first compared MinecraftOcc with the NYUv2 and OccScanNet datasets in terms of scene count, image count, resolution, and image quality as shown in Figure 1. To ensure an objective evaluation, we randomly selected 100 images from each dataset to assess image quality using the NIQE, PIQE, and LV metrics. For the occupancy prediction experiments, and to ensure a fair comparison with NYUv2 and OccScanNet, we mapped the 1,452 classes in MinecraftOcc to their 13 corresponding categories (see Appendix G for the detailed mapping). We initially evaluated several methods on our dataset at three scales (8k, 50k, and 100k), including MonoScene (Cao & De Charette, 2022), NDCScene(Yao et al., 2023), ISO(Yu et al., 2024), and Symphonies (Jiang et al., 2024). Additionally, we conducted joint training experiments combining Symphonies with NYUv2 and the 8k-scale MinecraftOcc dataset.

**Analysis.** Experimental results demonstrate that our MinecraftOcc dataset significantly outperforms NYUv2 and OccScanNet in terms of image quantity and metrics including NIQE, PIQE, and LV as shown in Table 3. As the results in Table 4, existing mainstream methods generally achieve relatively low performance, indicating the unique challenges presented by our dataset.

Table 3: Image quality comparison across datasets using NIQE(↓), PIQE(↓), and LV(↑).

| Dataset | NIQE ↓ | PIQE ↓ | LV ↑ |
|---|---|---|---|
| NYUv2 | 14.96 | 47.40 | 57,369 |
| OccScanNet | 17.63 | 58.78 | 10,352 |
| MinecraftOcc | **9.97** | **45.23** | **274,305** |

Notably, MonoScene, which delivers average performance on NYUv2, demonstrates remarkable stability on MinecraftOcc. We hypothesize that this is because other methods are overfitted to the NYUv2 dataset, leading to performance degradation when evaluated on a different dataset. Furthermore, after joint training with the NYUv2 dataset on Symphonies(Jiang et al., 2024), improvements in both IoU(**+0.43**) and mIoU(**+0.21**) were observed, suggesting that the MinecraftOcc dataset can effectively complement existing datasets, which is in Table 5.

Table 4: Minecraftocc Dataset results under different training settings.

| Setting | Method | IoU | mIoU | Precision | Recall | empty | ceiling | floor | wall | window | chair | bed | sofa | table | tvs | furniture | objects |
|---|---|---|---|---|---|---|---|---|---|---|---|---|---|---|---|---|---|
| 8k | Monoscene | **40.66** | 20.93 | 48.54 | 71.46 | **89.10** | **54.28** | 78.68 | **32.14** | 0.00 | 11.20 | 12.56 | 13.15 | 8.67 | 1.94 | 10.86 | 6.74 |
| | NDC-Scene | 37.06 | 17.82 | 46.51 | 65.57 | 88.42 | 46.01 | **79.10** | 28.03 | 0.00 | 9.37 | 3.26 | 5.81 | 11.86 | 0.81 | 8.18 | 3.61 |
| | ISO | 33.82 | 14.83 | 37.16 | **79.00** | 83.14 | 48.42 | 78.33 | 27.08 | 0.00 | 1.30 | 0.13 | 2.49 | 1.44 | 0.40 | 3.48 | 0.07 |
| | Symphonies | 39.11 | **21.56** | **49.30** | 65.42 | 89.05 | 49.39 | 77.16 | 31.06 | **2.80** | **12.24** | **13.27** | **13.26** | **13.77** | **4.95** | **11.11** | **8.13** |
| 50k | Monoscene | **39.51** | **19.61** | **54.00** | 59.55 | **91.37** | **52.93** | 83.99 | **27.68** | 2.57 | **9.70** | 5.82 | 8.78 | 5.53 | 3.25 | **9.56** | **5.90** |
| | NDC-Scene | 37.97 | 19.45 | 48.81 | 63.10 | 90.52 | 49.87 | 83.49 | 26.98 | **5.77** | **10.38** | 6.29 | **10.21** | 4.96 | 3.37 | 6.88 | 5.72 |
| | ISO | 35.07 | 15.69 | 42.06 | **67.82** | 88.22 | 44.46 | 80.81 | 25.40 | 0.56 | 2.98 | 3.07 | 4.05 | 3.50 | 1.17 | 5.02 | 1.55 |
| | Symphonies | 37.28 | **19.61** | 51.17 | 57.88 | 90.95 | 50.93 | **84.43** | 25.58 | 5.12 | 8.08 | **7.00** | 8.66 | **6.48** | **5.14** | 9.01 | 5.32 |
| 100k | Monoscene | **29.23** | **14.56** | 50.39 | 41.03 | **90.89** | 24.39 | 52.85 | **24.42** | 2.40 | **9.73** | 10.45 | 8.62 | 7.48 | 7.86 | 6.46 | 5.53 |
| | NDC-Scene | 28.08 | 12.96 | 41.09 | **47.02** | 88.91 | **26.12** | 52.81 | 22.83 | 1.11 | 6.93 | 8.26 | 6.86 | 6.41 | 2.76 | 4.61 | 3.85 |
| | ISO | 23.20 | 8.15 | 42.35 | 33.90 | 89.66 | 20.79 | 49.18 | 18.80 | 0.00 | 0.00 | 0.00 | 0.00 | 0.00 | 0.00 | 0.09 | 0.00 |
| | Symphonies | 27.60 | 13.08 | 34.78 | 57.23 | 86.14 | 26.05 | 61.02 | 21.27 | **2.97** | 6.06 | 5.90 | 5.74 | 4.79 | 2.50 | 4.53 | 3.05 |

Table 5: Performance comparison on NYU V2 Dataset. * Represents the model trained on a mixture of the MinecraftOcc 8k and NYUv2 training sets, and evaluated on the NYUv2 test set.

| Method | IoU | mIoU | ceiling | floor | wall | window | chair | bed | sofa | table | tvs | furniture | objects |
|---|---|---|---|---|---|---|---|---|---|---|---|---|---|
| LMSCNet | 33.93 | 15.88 | 4.49 | 88.41 | 4.63 | 0.25 | 3.94 | 32.03 | 15.44 | 6.57 | 0.02 | 14.51 | 4.39 |
| AICNet | 30.03 | 18.15 | 7.58 | 82.97 | 9.15 | 0.05 | 6.93 | 35.87 | 22.92 | 11.11 | 0.71 | 15.90 | 4.56 |
| 3DSketch | 38.64 | 22.91 | 8.53 | 90.45 | 9.94 | 5.67 | 10.64 | 42.29 | 29.21 | 13.88 | 9.38 | 23.83 | 8.19 |
| Monoscene | 42.51 | 26.94 | 8.89 | 93.50 | 12.06 | 13.57 | 13.72 | 48.19 | 36.11 | 15.13 | 15.22 | 27.96 | 12.94 |
| NDC-Scene | 44.17 | 29.03 | 12.02 | 93.51 | 13.11 | 13.77 | 15.83 | 49.57 | **39.87** | 17.17 | **24.57** | 31.00 | 14.96 |
| Symphonies | 49.91 | 29.70 | **14.54** | 86.59 | 25.95 | 15.96 | 16.78 | 46.60 | 38.06 | 15.37 | 15.32 | 32.16 | **19.58** |
| Symphonies* | **50.34** | **29.91** | 13.96 | **88.55** | **26.18** | **17.26** | **17.22** | 45.83 | 38.94 | **17.38** | 12.29 | **32.58** | 18.88 |

### 4.5 Comparison with Layout-Based Reconstruction Methods

**Implementation Details.** To validate the advantages of our occupancy-based reconstruction over layout-based methods, we compare ***World2Minecraft*** with the recent indoor scene generation methods: LayoutGPT (Feng et al., 2023), I-Design (Çelen et al., 2024), and LayoutVLM (Sun et al., 2025b). These methods generate scenes from textual descriptions but lack the geometric precision required for embodied AI tasks. We adapt them to our real-to-sim setting by converting input im-

Table 6: Comparison with layout-based scene generation methods.

| Method | OOB ↓ | Collision ↓ | Semantic ↑ | Visual ↑ | Complete ↑ | Aesthetic ↑ |
|---|---|---|---|---|---|---|
| LayoutGPT | 0.279 | 4.5 | 0.689 | 5.000 | 3.856 | 4.582 |
| I-Design | 0.423 | **0** | 0.884 | 6.001 | 4.734 | 5.352 |
| LayoutVLM | **0** | 0.9 | 0.348 | 3.625 | 2.270 | 2.708 |
| **World2Minecraft (Ours)** | 0.024 | 0.2 | **0.913** | **6.145** | **5.186** | **6.022** |

Table 7: Detailed efficiency breakdown comparing our pipeline with building from scratch.

| Metric Details | Build from Scratch | World2Minecraft (Ours) | Improvement |
|---|---|---|---|
| **Total Time (seconds)**↓ | 482.00 | **70.38** | – |
|   Automated Process | – | 5.88 | – |
|   Manual Refinement | 482.00 | 64.50 | 7.5× |
| **Total Operations**↓ | 340.00 | **24.50** | 13.9× |
|   Addition Actions | 319.30 | 9.70 | 32.9× |
|   Deletion Actions | – | 7.60 | – |
|   Orientation Adjustments | 20.70 | 7.20 | 2.9× |

ages into textual descriptions using GPT-4o, which are then used to generate scene layouts. For a fair comparison, we use the same set of scenes from the MinecraftVLN dataset that were manually refined in our *World2Minecraft* evaluation. Our evaluation employs six metrics spanning functionality and aesthetics: OOB Rate (percentage of objects placed outside room boundaries), Collision Count (number of intersecting objects), Semantic Integrity (the ratio of generated semantic categories to the total categories present in the ground truth scene), and Visual Realism, Scene Completeness, and Aesthetic Atmosphere (perceptual scores rated by GPT-4o on a scale of 1-10 assessing realism, completeness, and aesthetic appeal, respectively).

**Analysis.** As shown in Table 6, *World2Minecraft* outperforms baselines across most metrics, notably achieving 0.913 in *Semantic Integrity* and 6.145 in *Visual Realism*. The minimal OOB Rate (0.024) and Collision Count (0.2) reflect superior spatial awareness, whereas layout-based methods (e.g., LayoutGPT, LayoutVLM) struggle with geometric conflicts and plausibility. This success stems from the combination of semantic occupancy prediction and shape-aware template matching, which captures fine-grained geometry to ensure the precise alignment vital for avoiding obstacles in downstream VLN tasks.

### 4.6 EFFICIENCY ANALYSIS OF MANUAL REFINEMENT

**Implementation Details.** We conducted efficiency experiment comparing *World2Minecraft* with refinement against building scenes entirely from scratch. For 15 scenes from the MinecraftVLN-Base dataset, we measured the total time and operation counts. Experienced builders created equivalent scenes from scratch as a baseline. The manual refinement involves three simple, lightweight operations: **Deletion** of floating artifacts, **Completion** of minor surface holes, and **Adjustment** of object orientations.

**Results.** As shown in Table 7, *World2Minecraft* with refinement requires only **70.38s per scene**—a **7× reduction** compared to building from scratch (482.00s). The refinement process itself involves an average of **24.5 operations** (e.g., 9.7 hole fillings, 7.6 noise deletions, 7.2 orientation adjustments), compared to 340 operations for complete scene construction. This efficiency stems from our pipeline providing a high-quality initial reconstruction, requiring only minimal corrections. These corrections address imperfections inherent to any reconstruction algorithm, ensuring navigability for VLN tasks.

### 5 CONCLUSION

In this work, we introduce *World2Minecraft*, a framework that converts real-world scenes into structured Minecraft environments via 3D semantic occupancy prediction. We also propose scalable **data construction pipeline** and we build *MinecraftOcc*, a large-scale dataset of diverse indoor scenes with voxel-wise semantic annotations. Our experiments demonstrate the utility of these reconstructed environments for downstream VLN tasks and establish *MinecraftOcc*'s dual value as both a challenging new benchmark that exposes limitations in state-of-the-art models, and as a powerful resource for augmenting existing real-world datasets. To foster reproducibility and future research, we will publicly release our complete framework and dataset.

ETHICS STATEMENT

Our work aims to advance embodied AI in an ethical way, focusing on safety, transparency, and repeatability. All data in this study are artificially created inside the Minecraft virtual world, which offers a controlled and consistent experimental setup. The *MinecraftOcc* and *MinecraftVLN* datasets are built entirely from simulated scenes and include no real human data, private details, or identifiable personal spaces. This approach avoids privacy issues that come with collecting real-world data. By working mainly in simulation, we also lower the safety risks and resource use typically involved in real robot experiments.

REPRODUCIBILITY STATEMENT

To reproduce the work presented in this paper, follow these steps in sequence:

1. Download the **World2Minecraft** code along with the **MinecraftOcc** and **MinecraftVLN** datasets.
2. Train the **EmbodiedOcc** model using the provided configuration and training scripts.
3. Feed the predictions from the trained EmbodiedOcc model into the **World2Minecraft** pipeline to generate corresponding Minecraft construction commands.
4. Prepare a Minecraft environment with the **TMEO Mod** installed and execute the generated commands to reconstruct the scenes.
5. Conduct Vision-Language Navigation (VLN) tasks using the provided evaluation scripts within the reconstructed Minecraft scenes.
6. Collect image sequences using the **Screenshot with Coordinates** tool for data acquisition purposes.
7. Extract map data using the **WorldEdit** utility to obtain scene layout information.
8. Generate occupancy data using the provided processing scripts to create the final dataset format.

ACKNOWLEDGMENTS

This work was supported by the National Natural Science Foundation of China (Grant Nos. 62302167, 62222602, 62502159, U23A20343, W2521174), the Shanghai Committee of Science and Technology (Grant Nos. 25511103300, 25511104302, 25511102700), the Natural Science Foundation of Shanghai (Grant No. 25ZR1402135), the Natural Science Foundation of Chongqing (Grant Nos. CSTB2023NSCQ-JQX0007, CSTB2023NSCQ-MSX0137, CSTB2025NSCQ-GPX0445), the Young Elite Scientists Sponsorship Program by CAST YESS20240780, and the Open Project Program of the State Key Laboratory of CAD&CG, Zhejiang University (Grant No. A2501).

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

## A   THE USE OF LARGE LANGUAGE MODELS

In the preparation of this manuscript, large language models (LLMs) were utilized solely for writing and presentation assistance, in accordance with their respective licenses and terms of use. Specifically, LLMs were employed to assist in language polishing to improve the fluency and clarity of selected sections, adjusting the layout and presentation of figures and tables, and generating auxiliary code snippets for data processing and visualization tasks. It is important to emphasize that all scientific content, methodological development, experimental design, data analysis, and conclusions remain the intellectual contribution of the authors, with LLM usage strictly limited to auxiliary editorial and presentational tasks.

## B   PRELIMINARIES

Minecraft, as an open-world sandbox game, is renowned for its highly flexible building mechanics and consistent physics, making it a significant foundation for constructing simulation environments in embodied AI research. This section systematically introduces its standard environment features, modding mechanism, and the high-fidelity extended environment we developed based on it.

### B.1   NAIVE MINECRAFT ENVIRONMENT

The standard Minecraft environment provides a voxel-based and spatially discretized world—the entire world is divided into 1×1×1 unit voxel blocks, each corresponding to approximately $1m^3$ in the real world. This environment not only exhibits geometric regularity but also supports realistic physical interactions, demonstrating a high degree of authenticity and simulation fidelity, especially in terms of lighting, gravity, and terrain dynamics, which closely mirror real-world physical laws.

However, the vanilla Minecraft environment has notable limitations in embodied AI research. Its low-resolution, blocky visual style introduces a significant domain gap when compared to real-world images, restricting its applicability in perception tasks. Furthermore, the default Minecraft blocks lack diversity and granularity, consisting mainly of abstract classes (e.g., "wood", "stone") rather than semantically meaningful entities such as furniture. The indoor structures in the default environment are overly simplistic and lack layout complexity, failing to provide the spatial variety necessary for perception-dependent decision-making in embodied tasks.

### B.2   MOD ABOUT MINECRAFT

In order to address the limitations described above, we introduced a series of functional mods that significantly enhance environment construction, data collection, and visual realism. The modding system is a core extensibility mechanism of Minecraft, allowing modification or enhancement of game functionality through custom code and resource packs. In this work, we mainly employed three mods, as illustrated in Figure 3, namely:

**WorldEdit** It provides efficient large-scale procedural scene generation and editing capabilities. It supports rapid creation, duplication, and modification of composite structures via scripting, greatly improving both the efficiency and diversity of constructing complex indoor environments. Additionally, it allows easy access to the map data of target regions, including block types and coordinates, subsequent downstream processing and analysis.

**Screen with Coordinates** It simultaneously records the player's current viewpoint frame along with the player's position coordinates and viewing orientation, represented in Euler angles, during rendering.

**TMEO Texture and Mod Pack** It introduces over 1,400 fine-grained, semantically labeled object models (e.g., furniture such as "Blingds lighting" and household items like "Crib infant beds"), substantially enriching the semantic diversity and object hierarchy of scenes. Coupled with its high-resolution physically-based material pack, it lays the foundation for subsequent Physically Based Rendering(PBR).

### B.3 Extended Minecraft Environment

Building upon the aforementioned mods, we constructed a high-fidelity, diversified extended Minecraft environment tailored for embodied AI tasks. This environment significantly surpasses the native platform in terms of semantic complexity, visual realism, and scalability.

By integrating the fine-grained objects and diverse structures from the TMEO mod, we developed an indoor environment system comprising 156 detailed scenes and approximately 1,600 rooms. These encompass multi-story architectural structures, complex spatial layouts, and dense furnishings, greatly enriching the semantic hierarchy and spatial diversity of the scenes.

On the visual level, leveraging high-definition textures, PBR shaders, and dynamic lighting mods, we achieved realistic shadows, reflections, and global illumination effects. This markedly reduces the domain gap between synthetic images and real-world scenes.

In terms of environment construction and data collection, tools like WorldEdit and Screen with Coordinates enabled the establishment of a standardized pipeline. This supports the efficient generation of new scenes and the automatic acquisition of multi-modal annotated data, ensuring high reusability and extensibility.

---

**Algorithm 1** World2Minecraft: Reality-to-Virtual Transfer

---

**Require: Input:** Image set $\mathcal{I} = \{I_1, \ldots, I_N\}$
  1:     Camera intrinsic parameters $\mathcal{K}$
  2:     Camera extrinsic parameters $\mathcal{E}$
  3:     Pretrained models $\mathcal{F}_{\text{mono}}, \mathcal{F}_{\text{emb}}$
**Ensure: Output:** Reconstructed Minecraft scene
  4:  **procedure** RECONSTRUCTSCENE($\mathcal{I}, \mathcal{K}, \mathcal{E}, \mathcal{F}_{\text{mono}}, \mathcal{F}_{\text{emb}}$)
  5:     $\mathcal{O}_{\text{mono}} \leftarrow \emptyset$         ▷ Initialize monocular predictions set
  6:     **for** each image $I_i \in \mathcal{I}$ **do**       ▷ Process each view
  7:        $\mathcal{O}^i_{\text{mono}} \leftarrow \mathcal{F}_{\text{mono}}(I_i, \mathcal{K})$      ▷ Generate per-view occupancy
  8:        $\mathcal{O}_{\text{mono}} \leftarrow \mathcal{O}_{\text{mono}} \cup \{\mathcal{O}^i_{\text{mono}}\}$
  9:     $\hat{\mathcal{O}}_{\text{scene}} \leftarrow \mathcal{F}_{\text{embodied}}(\mathcal{O}_{\text{mono}}, \mathcal{K}, \mathcal{E})$   ▷ Fuse multi-view predictions
10:     $\mathcal{D} \leftarrow \mathbf{K} * \hat{\mathcal{O}}_{\text{scene}}$     ▷ Compute density map via 3D convolution
11:     $\mathcal{C} \leftarrow \{\mathbf{v} \in \hat{\mathcal{O}}_{\text{scene}} \mid \mathcal{D}(\mathbf{v}) \geq \tau\}$   ▷ Extract centers above threshold $\tau$
12:     $\mathcal{C}' \leftarrow \text{Cluster}(\mathcal{C}, \eta)$     ▷ Merge centers within distance $\eta$
13:     $\mathcal{M} \leftarrow \text{TranslateToMinecraft}(\mathcal{C}')$   ▷ Generate Minecraft building commands
14:     ExecuteCommands($\mathcal{M}$)     ▷ Render scene in Minecraft
15:     **return** MinecraftScene     ▷ Return reconstructed virtual scene

---

## C Algorithm of World2Minecraft

Our proposed method, World2Minecraft, formulates the reality-to-virtual transfer as a scene reconstruction problem. The core pipeline, outlined in Algorithm 1, takes a set of multi-view images and camera parameters as input and produces executable commands to reconstruct the scene in Minecraft. The process consists of three main stages: multi-view semantic occupancy prediction, volumetric fusion and density-based filtering, and finally, virtual world generation.

**Stage 1: Multi-view Semantic Occupancy Prediction.** The algorithm begins by processing each input image $I_i$ independently using a monocular prediction model $\mathcal{F}_{\text{mono}}$ (Line 4-7). This model infers an initial 3D semantic occupancy volume $\mathcal{O}^i_{\text{mono}}$ for each view, capturing the geometry and semantics visible from that particular viewpoint. These per-view predictions are aggregated into a set $\mathcal{O}_{\text{mono}}$.

**Stage 2: Volumetric Fusion and Filtering.** The individual occupancy volumes are then fused into a consistent global scene representation $\hat{\mathcal{O}}_{\text{scene}}$ by an embodied model $\mathcal{F}_{\text{embodied}}$, which utilizes the camera parameters to resolve inconsistencies and merge information (Line 8). To obtain a clean and structured representation suitable for building, we compute a density map $\mathcal{D}$ by applying a 3D convolution kernel $\mathbf{K}$ to the fused occupancy (Line 9). Voxel centers with a density value exceeding a threshold $\tau$ are selected as candidate building locations $\mathcal{C}$ (Line 10). A clustering step (Line 11)

further refines these candidates by merging those within a small distance $\eta$, reducing redundancy and ensuring structural coherence.

**Stage 3: Virtual World Generation.** The final stage translates the refined 3D centers $\mathcal{C}'$ into a sequence of Minecraft building commands $\mathcal{M}$ (Line 12). These commands, which specify the placement of specific block types at 3D coordinates, are executed to render the final scene within the Minecraft environment (Line 13), completing the transfer from reality to a semantically decomposed and editable virtual world.

## D IMAGE AND CAMERA PARAMETER ACQUISITION

As shown in Figure 3, we use an automated mod tool, *Screen with Coordinates*, to acquire data. This tool simultaneously captures first-person screenshots while recording the virtual camera's pose for each frame, including its 3D position and orientation (Euler angles). From this paired data, we compute the intrinsic and extrinsic matrices of the camera for each image.

### D.1 INTRINSIC CAMERA MATRIX

The intrinsic matrix $\mathcal{K}$ is determined by the virtual camera's Field of View (FOV) and the image dimensions $(W, H)$.

$$\mathcal{K} = \begin{bmatrix} f_x & 0 & c_x \\ 0 & f_y & c_y \\ 0 & 0 & 1 \end{bmatrix} \tag{10}$$

Here, the focal lengths $f_x, f_y$ and the principal point $(c_x, c_y)$ are derived from the horizontal FOV, denoted as $\alpha$:

$$f_x = f_y = \frac{W}{2 \tan(\alpha/2)}, \quad c_x = \frac{W}{2}, \quad c_y = \frac{H}{2} \tag{11}$$

### D.2 EXTRINSIC CAMERA MATRIX

The extrinsic matrix $\mathcal{E}$ defines the transformation from the camera coordinate system to the world frame. It is constructed from the camera's position $\mathbf{p} = (x_p, y_p, z_p)^T$ and its orientation, which is defined by a rotation matrix $\mathbf{R}$.

$$\mathcal{E} = \begin{bmatrix} \mathbf{R} & \mathbf{p} \\ \mathbf{0}^T & 1 \end{bmatrix} \tag{12}$$

The rotation matrix $\mathbf{R}$ is derived from the yaw ($\theta$) and pitch ($\phi$) angles provided by the game mod. To align the mod's Euler angle convention with a standard right-handed coordinate system (e.g., camera: +X right, +Y down, +Z forward), we apply an offset of $\pi$ to the angles.

The final orientation is achieved by composing two sequential rotations: first, a yaw rotation around the world's Y-axis, followed by a pitch rotation around the camera's local X-axis. This corresponds to an extrinsic YX Euler angle convention.

First, the yaw rotation matrix $\mathbf{R}_{\text{yaw}}$ is defined as a rotation around the Y-axis by an angle of $(\theta + \pi)$:

$$\mathbf{R}_{\text{yaw}} = \mathbf{R}_Y(\theta + \pi) = \begin{bmatrix} \cos(\theta + \pi) & 0 & \sin(\theta + \pi) \\ 0 & 1 & 0 \\ -\sin(\theta + \pi) & 0 & \cos(\theta + \pi) \end{bmatrix} = \begin{bmatrix} -\cos\theta & 0 & -\sin\theta \\ 0 & 1 & 0 \\ \sin\theta & 0 & -\cos\theta \end{bmatrix} \tag{13}$$

Next, the pitch rotation matrix $\mathbf{R}_{\text{pitch}}$ is defined as a rotation around the X-axis by an angle of $(\phi + \pi)$:

$$\mathbf{R}_{\text{pitch}} = \mathbf{R}_X(\phi + \pi) = \begin{bmatrix} 1 & 0 & 0 \\ 0 & \cos(\phi + \pi) & -\sin(\phi + \pi) \\ 0 & \sin(\phi + \pi) & \cos(\phi + \pi) \end{bmatrix} = \begin{bmatrix} 1 & 0 & 0 \\ 0 & -\cos\phi & \sin\phi \\ 0 & -\sin\phi & -\cos\phi \end{bmatrix} \tag{14}$$

The final rotation matrix $\mathbf{R}$ is the product of these two matrices, with the yaw rotation applied first:

$$\mathbf{R} = \mathbf{R}_{\text{pitch}} \cdot \mathbf{R}_{\text{yaw}} = \begin{bmatrix} -\cos\theta & 0 & -\sin\theta \\ \sin\theta\sin\phi & -\cos\phi & -\cos\theta\sin\phi \\ -\sin\theta\cos\phi & -\sin\phi & \cos\theta\cos\phi \end{bmatrix} \tag{15}$$

# E    VIEWPOINT PROJECTION AND REDUNDANCY REMOVAL

To further improve label quality, we introduce a **view frustum culling** mechanism based on camera projection. In early versions of the dataset, the occupancy regions associated with an image occasionally included irrelevant space outside the camera's field of view, which introduces noise. Therefore, we use the camera's intrinsic and extrinsic parameters to project the 3D voxel grid onto the 2D image plane. Let $\pi : \mathbb{R}^3 \to \mathbb{R}^2$ be the camera projection function, which utilizes the intrinsic $\mathcal{K}$ and extrinsic $\mathcal{E}$ matrices to map a world coordinate point $\mathbf{v}$ to image coordinates $\mathbf{u}$. We retain only those voxels that project within the image boundaries. The final set of visible voxels, $\mathcal{V}_{\text{visible}}$, is defined as:

$$\mathcal{V}_{\text{visible}} = \{\mathbf{v} \in \mathcal{V} \mid \pi(\mathbf{v}, \mathcal{E}, \mathcal{K}) \in [0, W] \times [0, H]\} \tag{16}$$

By removing voxels that project outside the image, we ensure that the final 3D occupancy labels are precisely aligned with the image's field of view, thereby eliminating redundancy and preventing label misalignment.

# F    DATASET ANALYSIS

Table 8: Statistical Summary of Instruction Lengths Across Different Tasks and Datasets. The table shows the count, mean, standard deviation (Std. Dev.), and quartiles for the length of navigation instructions.

| Task | Dataset Type | Count | Mean | Std. Dev. | Min | 25% | 50% | 75% | Max |
|------|-------------|-------|------|-----------|-----|-----|-----|-----|-----|
| Next-Action | Base | 769 | 79.01 | 32.87 | 15 | 57.0 | 78.0 | 96.0 | 187 |
| | Extend | 1351 | 171.37 | 67.10 | 62 | 117.0 | 156.0 | 232.0 | 334 |
| | Combined | 2120 | 137.87 | 72.34 | 15 | 81.0 | 121.0 | 181.0 | 334 |
| Next-View | Base | 290 | 87.41 | 33.57 | 28 | 64.0 | 87.0 | 99.0 | 187 |
| | Extend | 1132 | 173.23 | 66.70 | 62 | 121.0 | 156.0 | 232.0 | 334 |
| | Combined | 1422 | 155.73 | 70.48 | 28 | 103.0 | 145.0 | 208.0 | 334 |

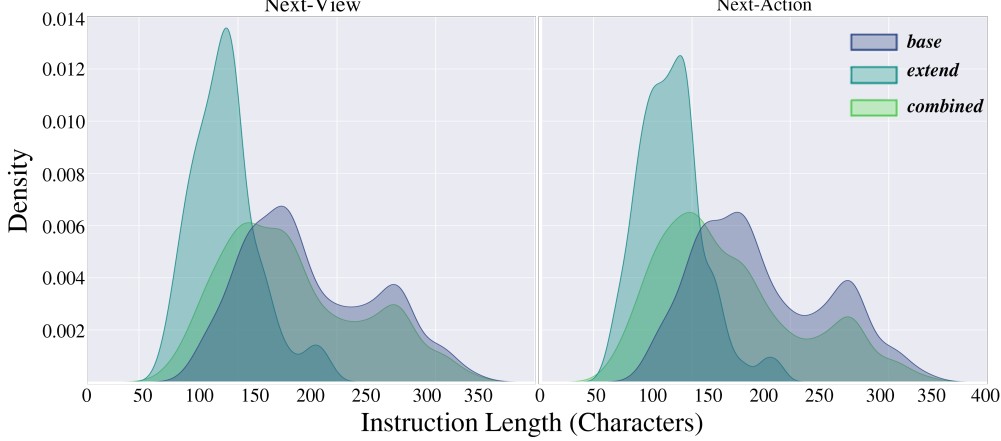

Figure 6: A comparison of instruction length distributions across three datasets for the Next-View and Next-Action tasks. The extend dataset clearly contains shorter and more uniform instructions.

# G    SEMANTIC CLASS MAPPING FOR CROSS-DATASET EXPERIMENTS

To facilitate a fair and meaningful comparison between models trained on our `MinecraftOcc` dataset and those evaluated on standard benchmarks like NYUv2, we established a many-to-one

Table 9: Dataset statistics for Minecraftocc and MinecraftVLN. Minecraftocc is provided at three scales (8k, 50k, 100k), while MinecraftVLN includes three scales (Base, Extend, Combined) with two task settings: Next-View and Next-Action.

| Dataset | Scale | Task | Train | Test | All |
|---------|-------|------|-------|------|-----|
| Minecraftocc | 8k | – | 6,100 | 2,024 | 8,124 |
| | 50k | – | 39,914 | 10,245 | 50,159 |
| | 100k | – | 79,799 | 20,366 | 100,165 |
| MinecraftVLN | Base | Next-View | 902 | 230 | 1132 |
| | | Next-Action | 1042 | 309 | 1351 |
| | Extend | Next-View | 249 | 41 | 290 |
| | | Next-Action | 594 | 175 | 769 |
| | Combined | Next-View | 1151 | 271 | 1422 |
| | | Next-Action | 1636 | 484 | 2120 |

mapping from our 1,000+ fine-grained, Minecraft-specific classes (including 200 distinct lighting fixtures) to a standardized set of 12 common indoor semantic categories. This process creates a shared semantic ground for consistent evaluation.

The mapping was designed to group Minecraft blocks and items based on their functional and structural roles within an indoor scene. Table 10 summarizes this hierarchy, providing the rationale for each target superclass along with a few representative examples from the `MinecraftOcc` dataset. This standardized taxonomy is used for all cross-dataset experiments.

The complete, exhaustive mapping of all classes is provided as a `.json` file in our supplementary material to ensure full reproducibility.

Table 10: Summary of the mapping from our fine-grained `MinecraftOcc` classes to 12 target superclasses. Representative examples are provided for clarity.

| Target Superclass | Mapping Rationale / Included Concepts | Example `MinecraftOcc` Classes |
|-------------------|---------------------------------------|-------------------------------|
| empty | A broad catch-all category for non-structural, transparent, or empty elements. | `tmeo_ultra:chuanglian...` `tmeo_ultra:diaodeng...` |
| ceiling | Overhead structural surfaces and decorative ceiling elements. | `tmeo_ultra:shigaoxian...` `minecraft:birch_planks` |
| floor | Horizontal walking surfaces, including stairs, slabs, and floor coverings. | `tmeo_ultra:yitishilouti...` `minecraft:stone_brick_slab` |
| wall | Core vertical structural surfaces | `tmeov:diantimen` |
| window | All types of blinds and shutters. | `tmeov:baiyechuang` |
| chair | All single-person seating, including benches, stools, and dining chairs. | `tmeo_ultra:changdeng_1x_3` `tmeo_ultra:bataiyizi` |
| bed | All types of beds and associated bedding parts. | `tmeo_ultra:dachuangban...` |
| sofa | Couches and other seating furniture. | `tmeo_ultra:shafa_1x_2` |
| table | Surfaces for placing objects. | `tmeo_ultra:canzhuoyuanxing` |
| tvs | All television sets and monitor screens. | `tmeo_ultra:diaoguadianshi` |
| furn | General non-seating furniture, e.g., cabinets, shelves, sinks, and wardrobes. | `tmeo_ultra:yigui...` `tmeov:shujia...` |
| objs | Miscellaneous functional and decorative items not part of the main structure. | `tmeov:penzai` `tmeo_ultra:yinxiang...` |

## H  IMAGE QUALITY METRICS

This appendix briefly describes the three no-reference image quality assessment metrics used in our evaluation:

**Natural Image Quality Evaluator(NIQE).** Measures how closely an image's statistical properties match those of natural images. Lower values indicate better, more natural quality.

**Perception-based Image Quality Evaluator(PIQE).** Assesses local distortions and artifacts in images based on human visual perception. Lower values indicate fewer distortions and better perceptual quality.

**Laplacian Variance(LV).** Quantifies image sharpness by measuring the variance of Laplacian-filtered responses. Higher values indicate sharper images with more detail.

These complementary metrics provide a comprehensive assessment of visual quality from different perspectives: NIQE evaluates global naturalness, PIQE detects local artifacts, and LV measures sharpness.

# I    IMPLEMENTATION DETAILS

This section provides a comprehensive overview of the implementation details for the two main experiments conducted in this study: Reinforcement Fine-Tuning (RFT) and Supervised Fine-Tuning (SFT). All experiments used `Qwen2.5-VL` as the base model.

## I.1    EXPERIMENT 1: REINFORCEMENT FINE-TUNING

The RFT experiment was conducted using the EasyR1 framework. For this experiment, the vision tower of the model was unfrozen and trained alongside the language components. We employed the Generalized Rejection Policy Optimization (GRPO) algorithm. To regularize the policy, we incorporated a Kullback-Leibler (KL) divergence penalty with a coefficient of $1.0 \times 10^{-2}$, calculated using a low-variance estimator.

**Dataset and Data Processing.** The training and validation data were sourced from Parquet files, using `content` for prompts and `answer` for responses. Prompts were formatted via a custom Jinja template (`mc.jinja`). The maximum prompt length was set to 4096 tokens, and the maximum response length was 1024 tokens.

**Hardware and Training Configuration.** The experiment was run on a single node with 8 GPUs, utilizing Fully Sharded Data Parallelism (FSDP) with full parameter sharding and CPU offloading for both model parameters and optimizer states to conserve memory. The rollout phase was accelerated with a tensor parallel size of 2. Key hyperparameters are summarized in Table 11.

## I.2    EXPERIMENT 2: SUPERVISED FINE-TUNING (SFT)

The SFT experiment was conducted using the LLaMA Factory framework.

**Model and Fine-tuning Strategy.** In contrast to the RFT experiment, the vision tower and the multimodal projector were kept frozen during SFT. Fine-tuning was performed only on the parameters of the language model component using a full-parameter approach (`finetuning_type: full`).

**Dataset and Preprocessing.** We utilized a custom dataset named `base_train_task1`, limiting the training to a maximum of 1000 samples. The data was formatted with the `qwen2_vl` conversation template, and the maximum sequence length was capped at 8192 tokens.

**Training Configuration.** The model was trained for 5 epochs using the DeepSpeed ZeRO Stage 3 strategy and `bfloat16` mixed precision. We employed a cosine learning rate scheduler with a 10% warmup period. No validation was performed during training. The detailed hyperparameters are presented in Table 11.

# J    INTERACTIVE VISUALIZATION TOOL : SCENEFORGE

To facilitate the analysis and refinement of occupancy clustering results, we developed an interactive web-based visualization tool names SceneForge that provides Open3D-like 3D exploration capabilities. This tool plays a crucial role in our *World2Minecraft* pipeline by enabling intuitive inspection and manual correction of semantic occupancy predictions.

Table 11: Comparison of key hyperparameters for the SFT and RFT experiments.

| Parameter | RFT Setting | SFT Setting |
|---|---|---|
| *General Strategy* | | |
| Fine-tuning Type | – | Full-parameter |
| Training Precision | – | bfloat16 |
| Algorithm | GRPO | – |
| *Optimization* | | |
| Optimizer | AdamW | AdamW (implied) |
| Learning Rate | $1.0 \times 10^{-6}$ | $1.0 \times 10^{-5}$ |
| Weight Decay | $1.0 \times 10^{-2}$ | – |
| LR Scheduler | – | Cosine |
| Warmup Ratio | – | 0.1 |
| Total Epochs | 30 | 5 |
| *Batching* | | |
| Global Batch Size | 128 | – |
| Per-device Batch Size | – | 1 |
| Gradient Accumulation Steps | – | 2 |
| Effective Batch Size | 128 (Global) | 2 |
| *RFT-Specific Details* | | |
| KL Coefficient ($\lambda_{\text{KL}}$) | $1.0 \times 10^{-2}$ | – |
| Rollout Samples ($n$) | 5 | – |
| Training Temperature | 1.0 | – |

## TOOL OVERVIEW

The visualization tool is implemented as a standalone web application using D3.js for 3D rendering and interaction. It supports the following key functionalities:

- **Multi-format Data Loading**: The tool accepts both voxel-wise occupancy data (`occ.json`) and pre-computed center points (`centers.json`), with optional demo data for quick testing.

- **Interactive 3D Exploration**: Users can rotate the view (mouse drag), zoom (scroll wheel), and pan (Shift + drag) to examine the scene from any angle, mimicking the interaction paradigm of Open3D.

- **Category-aware Visualization**: Twelve semantic categories are color-coded and can be individually toggled on/off via an interactive legend, enabling focused analysis of specific object types.

- **Real-time Parameter Adjustment**: Dynamic controls allow users to adjust voxel size, transparency, and center point size during visualization to optimize clarity.

- **Center Point Editing**: An advanced editing mode supports manual refinement of object centers through:
  - Drag-and-drop repositioning of center points
  - Point deletion (right-click) and selective removal
  - Point splitting for fine-grained object separation
  - Bulk operations by category selection

## J.1 INTEGRATION WITH RESEARCH WORKFLOW

The tool served two primary purposes in our research:

1. **Qualitative Analysis**: During method development, we used the tool to visually inspect clustering results, identify failure cases, and understand the limitations of automatic center prediction algorithms.

2. **Data Refinement**: For the *MinecraftVLN* dataset creation, the editing capabilities allowed us to manually correct inaccurately predicted object centers, ensuring higher quality navigation environments.

## J.2 Technical Implementation

The tool architecture consists of four modular components:

- `config.js` - Configuration constants and color schemes
- `data.js` - Data loading and processing utilities
- `projection.js` - 3D projection and rendering engine
- `ui.js` - User interface event handlers and state management

This web-based implementation ensures cross-platform compatibility without requiring complex dependencies, making it accessible for researchers to reproduce and extend our work. The complete source code is available in our supplementary materials.

