# OpenReview forum: "World2Minecraft: Occupancy-Driven Simulated Scenes Construction"
_ICLR.cc/2026/Conference — ICLR 2026 Poster_

### Official Review · Reviewer_X8ju · 2025-10-26

**Soundness:** 3
**Presentation:** 3
**Contribution:** 3
**Rating:** 4
**Confidence:** 3

**Summary:**

This paper introduces **World2Minecraft**, a framework for converting real-world scenes into Minecraft environments using 3D semantic occupancy prediction. Based on the reconstructed scenes, this paper also presents two datasets: **MinecraftOcc**, a large-scale benchmark for indoor 3D occupancy prediction, and **MinecraftVLN**, designed for Vision-Language Navigation tasks. Experiments demonstrate the framework’s scalability and effectiveness in improving model performance and enabling embodied AI research in editable virtual environments.

**Strengths:**

1. The paper is well-written, with clear and intuitive figures that make the dataset construction process easy to understand.

**Weaknesses:**

1. The use of occupancy as an intermediate representation for real-to-sim scene reconstruction raises some questions. According to the paper, occupancy is not directly used to reconstruct Minecraft scenes but instead to identify object instance centers for reconstruction. Does this suggest that object instance or room layout information alone is sufficient for scene reconstruction? Why not use these representations directly, considering they are potentially easier to obtain than occupancy labels?

2. Based on the results in Table 5, adding **MinecraftOcc** training data only provides a marginal improvement in model performance (mIoU +0.21). This raises concerns about the actual impact of **MinecraftOcc** in improving current occupancy prediction models or supplementing existing datasets.

3. The paper lacks experiments demonstrating the contribution of the **MinecraftVLN** dataset to existing visual navigation models or its ability to complement existing datasets. Additional experiments are recommended to show that **MinecraftVLN** can improve the performance of current visual navigation models.

**Questions:**

See weaknesses. If these concerns are clarified with solid results, I would be happy to increase my score.

---

> ### Author Response · Authors · 2025-11-23
> **Response to Reviewer X8ju**
>
> ***Part One***
>
> We sincerely thank the reviewer for their positive assessment of our writing and presentation, as well as their thoughtful questions which help clarify the core contributions of our work.
>
> ## **1. Response to Weakness 1: Why Occupancy instead of Layout?**
> ***1. Why not Only Layout?***
> This is an insightful question. We argue that room layout alone is insufficient for high-fidelity real-to-sim transfer. While layout can provide the basic structure of a scene (walls, floors), it fails to capture the detailed geometry, scale, and specific identity of individual objects. Our pipeline goes beyond structural layout. Relying solely on layout would lead to two fundamental problems that our occupancy-based approach directly solves:
> * **Shape & Scale Awareness:** Our pipeline fundamentally requires semantic occupancy, as structural layout alone is insufficient for realistic reconstruction. While a layout defines room boundaries, it **fails to capture the critical scale and shape of individual objects.** For instance, it cannot distinguish a single bed from a king-size bed—a vital difference that directly impacts navigability, as misplacing an oversized bed could block a virtual agent's path. Semantic occupancy provides the precise 3D geometry of each instance, enabling our shape-aware template matching to select library assets that **accurately reflect real-world scale and form**. This ensures the reconstructed environment is not only visually faithful but also functionally plausible for downstream tasks.
> * **Instance-Level Matching:** Because we reconstruct the occupancy first, we obtain the specific voxel shape of each instance. This allows our template matching to **select assets that match the *geometry* of the real-world object**, not just its semantic class. This provides a denser, more informative reconstruction than layout-only approaches.
>
> ***2. Comparison with Existing Methods Based On Layout***
>
> Unlike prior works that primarily rely on layout estimation or bounding boxes, our method provides a more geometrically accurate reconstruction by directly matching the predicted semantic occupancy against 3D furniture models. This occupancy-based template matching enables superior scale consistency and shape alignment, as validated in the following experiment. As shown in the results table, our approach achieves **the highest scores in Semantic Integrity (0.913) and Visual Realism (6.145)**, demonstrating its ability to preserve fine-grained geometric details from RGB images to the final scene—an end-to-end capability not achieved by existing methods.
> | Method | OOB Rate $\downarrow$ | Collision Count $\downarrow$ | Semantic Integrity $\uparrow$ | Visual Realism $\uparrow$ | Scene Completeness $\uparrow$ | Aesthetic Atmosphere $\uparrow$ |
> | :--- | :---: | :---: | :---: | :---: | :---: | :---: |
> | LayoutGPT | 0.279 | 4.5 | 0.689 | 5.000 | 3.856 | 4.582 |
> | I-Design | 0.423 | **0** | 0.884 | 6.001 | 4.734 | 5.352 |
> | LayoutVLM | **0** | 0.9 | 0.348 | 3.625 | 2.270 | 2.708 |
> | **World2Minecraft** | 0.024 | 0.2 | **0.913** | **6.145** | **5.186** | **6.022** |

---

> ### Author Response · Authors · 2025-11-23
>
> ***Part Two***
> ## **2. Response to Weakness 2: The Value of MinecraftOcc**
>
> Our contribution needs to be re-emphasized about MinecraftOcc, which extends **beyond the scale of the dataset**. While we provide a substantial benchmark in MinecraftOcc (over 100,000 images), we equally importantly introduce an **automated methodology for generating 3D semantic occupancy data**. This pipeline dramatically lowers the barrier for creating large-scale, high-quality 3D datasets, offering a practical solution to the field's critical data scarcity problem and enabling future community-driven data expansion.
>
> We appreciate the reviewer's attention to the quantitative results. While the absolute mIoU gain on NYUv2 in Table 5 is indeed modest (+0.21), this result carries significant meaning when properly contextualized:
>
> First, as supplementary training data, MinecraftOcc demonstrates clear practical value. The improvement on the real-world NYUv2 benchmark confirms that our synthetic data provides **complementary 3D structural knowledge** that enhances model robustness, despite the domain difference. This is particularly noteworthy given the inherent limitations of existing datasets - our image quality analysis below reveals that datasets like NYUv2 and OccScanNet suffer from substantial **noise and low resolution**, making mere incremental optimization on these test set of limited value for advancing general 3D understanding.
> | Dataset | NIQE (↓) | PIQE (↓) | LV (↑) |
> |---------|----------|----------|--------|
> | NYUv2 | 14.96 | 47.40 | 57,369 |
> | OccScanNet | 17.63 | 58.78 | 10,352 |
> | MinecraftOcc | 9.97 | 45.23 | 274,305 |
>
> *Table: Image quality comparison across datasets using NIQE(↓), PIQE(↓), and LV(↑).*
>
> More importantly, MinecraftOcc serves as a crucial new benchmark that reveals **fundamental limitations in current approaches**. As starkly demonstrated in Table 4, the dramatic performance drop of SOTA models exposes their severe overfitting to existing limited datasets. This diagnostic capability, **to test true generalization rather than within-distribution performance**, represents the core value of MinecraftOcc. By providing a large-scale, high-quality testbed with diverse scene layouts and lighting conditions, MinecraftOcc drives the field toward more generalizable 3D scene understanding beyond the constrained settings of current benchmarks.

---

> ### Author Response · Authors · 2025-11-23
>
> ***Part Three***
> ## **3. Response to Weakness 3: The Contribution of MinecraftVLN**
>
> We appreciate the reviewer's feedback. We wish to clarify that MinecraftVLN serves **primarily as a validation tool to demonstrate that our core pipeline produces functionally viable environments**. Its purpose is to confirm that agents can successfully perform embodied navigation tasks within our reconstructed scenes.
>
> MinecraftVLN should be viewed as a natural byproduct of our main workflow, rather than to the navigation task and dataset. The central innovations of this work remain:
>
> - The World2Minecraft real-to-sim pipeline
>
> - The automated methodology for generating large-scale 3D occupancy data (MinecraftOcc)
>
> The successful navigation results in MinecraftVLN **provide important evidence of our pipeline's practical utility**, while the dataset itself offers a useful resource for the community. However, we emphasize that our fundamental contributions lie in the scalable frameworks for environment reconstruction and dataset generation.

---

### Official Review · Reviewer_Nuc5 · 2025-10-28

**Soundness:** 2
**Presentation:** 2
**Contribution:** 2
**Rating:** 2
**Confidence:** 4

**Summary:**

This paper focuses on transforming real-world scenarios into Minecraft structured environments, attempting to address issues such as data pollution and insufficient flexibility in embodied intelligent simulation platforms.

This paper proposes a new VLN method **MinecraftVLN** that enables the agent to complete navigation tasks in the reconstructed scene.

This paper also proposes a large-scale dataset **MinecraftOcc** to benefit the entire community.

**Strengths:**

**The research question is important**: Existing simulation platforms (Habitat and AI2-THOR) are not editable, necessitating a real-to-sim approach. However, existing real-to-sim approaches lack physical properties and cannot be applied to downstream tasks. Therefore, the authors propose a solution based on 3D semantic occupancy prediction.

**Constructed two large datasets**: Data collection is a very tedious task. The authors proposes a low-cost collection pipeline to generate a large number of images, indoor scenes, and navigation samples. These have contributed to the community and navigation tasks, alleviating the previous problem of insufficient data.

**Weaknesses:**

**The method lacks innovation**: The "World2Minecraft framework" proposed in the article is essentially a patchwork of the "single-view 3D semantic occupancy prediction → multi-view fusion → Minecraft command generation" pipeline, with no groundbreaking design in any of the steps. The entire process lacks original technical support and is more like an "assembly experiment" of existing tools. The authors fail to explain the core issues of this pipeline, why previous researchers haven't developed this pipeline, whether the simple assembly of such a pipeline is directly applicable, whether additional problems were encountered during the process, and how the authors propose carefully designed modules to address these issues. I don't see a research-based process, but rather a collection of engineering pieces.

**The dataset's value may be limited**: I greatly appreciate the authors' meticulous effort in designing and collecting datasets for the benefit of the community and research. However, there are some concerns. The MinecraftOcc dataset has 156 scenes. While it contains "approximately 1,000 rooms," it doesn't specify the diversity between scenes (e.g., whether there is overlap in room layout, furniture layout, or lighting conditions), making its value in testing model generalization capabilities unproven.

**The introduction is poorly written**: When discussing the "flaws of existing methods," it first criticizes the "uneditability" of Habitat and AI2-THOR, then mentions the "lack of physical properties" of NeRF and 3D Gaussian splatting, and finally turns to semantic occupancy prediction methods. This leads to a logical jump and a failure to form a coherent chain of argumentation.

**The results are not very impressive**: The outcomes shown in Figure 4, Table 2, and Table 4 are not very impressive, and I did not perceive the advantages of the method and dataset proposed by the authors.

**Questions:**

Could the authors provide a detailed explanation of the core research question? What challenges did you encounter while addressing this issue that resulted in the final pipeline? I do not perceive the uniqueness of World2Minecraft, nor do I see any significant insights here.

Additionally, do the authors have better experimental results? In my opinion, the current experimental outcomes do not convincingly demonstrate the advantages of your method and dataset.

---

> ### Author Response · Authors · 2025-11-23
> **Response to Reviewer Nuc5**
>
> ***Part 1***
> ## **1. Response to Weakness 1 and Question 1: "the core research question" & "significant insights of World2Minecraft"**
>
> ***1. Significant Insights of Our Work***
> The reviewer questions the innovation of the World2Minecraft pipeline. We respectfully clarify that our work delivers **foundational contributions** through **two key innovations**.
> - To the best of our knowledge, we present **the first complete pipeline** that translates **single real-world images** into fully editable and interactive Minecraft environments, subsequently **enabling the training and testing of embodied AI agents** within them. This end-to-end realization from image input to functional embodied AI training environment is in itself a significant novel contribution.
> - Our contribution extends **beyond the scale of the dataset**. While we provide a substantial benchmark in MinecraftOcc (over 100,000 images), we equally importantly introduce an **automated methodology for generating 3D semantic occupancy data**. This pipeline dramatically lowers the barrier for creating large-scale, high-quality 3D datasets, offering a practical solution to the field's critical data scarcity problem and enabling future community-driven data expansion.
>
> ***2. The Core Research Question***
> Motivated by the scarcity of high-quality simulators, establishing an end-to-end framework from single images to **trainable agent environments is both pressing and promising**. To construct such a complete training pipeline, we propose World2Minecraft, which takes multi-view images as input, reconstructs semantic occupancy predictions of the entire scene, and after post-processing, builds an interactive and editable environment in Minecraft for agent training and evaluation. **This demonstrates the core research problem we aim to address and our proposed solution**. We believe our work identifies a genuine existing challenge and presents a novel, comprehensive framework to resolve it.
>
> ***3. Technical Challenges and Our Solutions:***
>  The process was far from a simple "assembly." We encountered and solved several critical challenges:
>
> - **Why Semantic Occupancy:** We selected semantic occupancy as our intermediate representation because it uniquely combines editable physical properties with proven occlusion handling capabilities—qualities that have made it successful in autonomous driving applications. This representation provides both the geometric completeness needed for reconstruction and the structural semantics required for meaningful scene editing.
> - **Geometric Discretization:** Converting continuous occupancy probabilities into discrete, stable Minecraft blocks required a novel convolutional filtering and thresholding approach to avoid noisy, floating structures.
>
> - **Semantic Asset Matching:** We developed a shape-aware template matching algorithm to align the predicted semantic occupancy of objects with 3D assets from our library, ensuring geometric fidelity beyond simple semantic label matching.
>
> - **Pose Estimation:** We implemented an optimization routine to estimate correct object orientations by comparing various rotated asset templates against the predicted occupancy.
> - **Embodied Agent Control:** We established a complete interaction framework between agents and Minecraft. The system provides agents with visual observations from the game environment and translates their action decisions into keyboard commands executed within Minecraft. This bidirectional interface enables seamless embodied task execution in the reconstructed scenes.
>
> These solutions, while perhaps not "groundbreaking" in isolation, were carefully designed and are essential for the pipeline's functionality. They represent a significant engineering and research effort to move from theory to a practical, automated system.

---

> ### Author Response · Authors · 2025-11-23
>
> ***Part 2***
> ## **(Additional)1. Response to Weakness 1 and Question 1: "the core research question" & "significant insights of World2Minecraft"**
> ***4. Why previous researchers haven't developed this pipeline***
>
> - **Interdisciplinary Integration Complexity** Constructing a complete workflow from images to interactive training environments presents extraordinary complexity spanning multiple research domains. While individual components have been studied in isolation, integrating them into a cohesive system requires expertise across computer vision, graphics, and embodied AI—a challenging interdisciplinary endeavor.
> - **Platform Selection Innovation** The platform selection dilemma posed a major obstacle. Even with successful scene reconstruction, rendering environments within a physics-grounded engine remained unresolved. Our key insight was identifying Minecraft as the ideal platform that provides both native physics simulation and unparalleled editability—a creative solution overlooked by previous work.
> - **Technological Readiness** Timing The underlying reconstruction technology has only recently matured to support such an ambitious pipeline. Earlier attempts were constrained by limitations in 3D understanding and semantic construction, making semantically-aware scene reconstruction practically infeasible until recent advances.
> - **Emerging Research Demand** The unprecedented surge in demand for embodied AI simulators, driven by breakthroughs in foundation models, has created an urgent need that didn't exist at this scale before. While timing is now optimal, the technical barriers we've overcome demonstrate why this pipeline represents meaningful advancement rather than an obvious combination of existing tools.
>
> ***5. Compare To Other Works***
> Unlike some contributions that focus on **highly specialized technical modules with limited scope** and **narrow applicability**, our work distinguishes itself through its **strong practical applicability** and comprehensive, end-to-end framework. While incremental module improvements often demonstrate value only within specific domains or experimental settings, our World2Minecraft pipeline delivers immediate practical utility for embodied AI research, simulation, and training. This represents a significant aspect of our contribution and addresses a crucial gap in current research infrastructure.

---

> ### Author Response · Authors · 2025-11-23
>
> ***Part 3***
> ## **2. Response to Weakness 4 and Question 2: No Impressive Results**
>
> We respectfully disagree that the results are unimpressive. They are designed to demonstrate specific, important points:
>
> - **Figure 4 (Reconstruction in Minecraft)** Our pipeline achieves high-fidelity reconstruction from real-world scenes to Minecraft environments across multiple views. The intermediate occupancy predictions (Prediction column) and final reconstructed scenes (Construction column) closely match both the input reality and predicted geometry, validating our method's geometric consistency and readiness for embodied AI tasks.
> - **Table 4 (Benchmark on MinecraftOcc)** The primary goal here is not to show a "better" method but to expose a critical generalization gap. The fact that SOTA models perform poorly on MinecraftOcc reveals their over-reliance on the limited priors of existing datasets (NYUv2/ScanNet). This is a valuable finding for the community.
> - **Table 5 (Joint Training)** This is a key result. Improving NYUv2 performance by adding MinecraftOcc data proves that our dataset provides complementary 3D knowledge, not just a domain shift. This demonstrates its value in learning more robust geometric representations.
> - **MinecraftVLN** The successful training of VLN agents (Table 2) validates the practical utility of our full pipeline. It proves that our reconstructed scenes are of sufficient quality and complexity to support meaningful downstream task evaluation.
> - **Comparison with the existing Methods** further validates our approach. As shown in the below table,  our approach achieves **the highest scores in Semantic Integrity (0.913) and Visual Realism (6.145)**, demonstrating its ability to preserve fine-grained geometric details from RGB images to the final scene—an end-to-end capability not achieved by existing methods.
>
> | Method | OOB Rate $\downarrow$ | Collision Count $\downarrow$ | Semantic Integrity $\uparrow$ | Visual Realism $\uparrow$ | Scene Completeness $\uparrow$ | Aesthetic Atmosphere $\uparrow$ |
> | :--- | :---: | :---: | :---: | :---: | :---: | :---: |
> | LayoutGPT | 0.279 | 4.5 | 0.689 | 5.000 | 3.856 | 4.582 |
> | I-Design | 0.423 | **0** | 0.884 | 6.001 | 4.734 | 5.352 |
> | LayoutVLM | **0** | 0.9 | 0.348 | 3.625 | 2.270 | 2.708 |
> | **World2Minecraft** | 0.024 | 0.2 | **0.913** | **6.145** | **5.186** | **6.022** |

---

> ### Author Response · Authors · 2025-11-23
>
> ***Part 4***
> ## **3. Response to Weakness 2: The Value of MinecraftOcc**
> Our contribution needs to be re-emphasized about MinecraftOcc: we provide **not only the MinecraftOcc** dataset itself, but more importantly, the **automated pipeline** for generating such 3D semantic occupancy data. While the dataset serves as an immediate, large-scale benchmark, the another core value lies in our methodology that enables efficient, scalable data creation - offering the community both a valuable resource and a sustainable solution to the fundamental challenge of data scarcity in 3D scene understanding.
>
> We agree that demonstrating dataset diversity is crucial. The MinecraftOcc dataset is built from **156 distinct, high-quality community-built scenes**, encompassing a wide variety of layouts and styles, comprising 156 distinct community-built scenes, MinecraftOcc includes: **living rooms, bedrooms, kitchens, bathrooms, basements, gyms, libraries, indoor pools, and large mansions (each containing over 20 rooms)**. Particularly noteworthy is our collection of **over 200 distinct lighting fixtures**, which create unique illumination conditions that further enhance the visual diversity and challenge models to generalize across varying lighting scenarios.This coverage spans both common domestic layouts and complex, spacious environments that are notably absent from existing benchmarks.

---

> > ### Comment · Reviewer_Nuc5 · 2025-11-27
> >
> > Thank you for your explanation regarding your contribution to the MinecraftOcc dataset. I agree with your emphasis not only on the dataset itself, but also on the automated process for generating 3D semantic occupancy data. This truly provides a sustainable solution to the data scarcity problem in 3D scene understanding.
> >
> > However, why is "200 distinct lighting fixtures" considered a significant contribution, especially since it's barely mentioned in the main text? And why does lighting have such a significant impact, when most methods perform similarly under different lighting conditions (i.e., changing the lighting doesn't significantly affect model performance)?
> >
> >
> > Also, the authors provided better experimental results, but I don't understand why they weren't included in the submission.
> >
> > **Overall, the authors' rebuttal resolved most of my concerns, and this paper could be revised even better.**
> >
> > I would be happy to raise my score to 6. Thank you.

---

> > > ### Author Response · Authors · 2025-11-28
> > >
> > > # Reply to the new questions
> > > We are deeply grateful to the reviewer for their positive reassessment of our work and rebuttle for raising the score. We are extremely encouraged that our clarifications were effective and that the value of our automated pipeline is recognized.
> > >
> > > Here are our responses to the additional questions:
> > >
> > > ## **1. Regarding the significance of the "200 distinct lighting fixtures"**
> > > The mention of diverse lighting was intended primarily as an **illustration of the broad categorical coverage and realism** of the MinecraftOcc dataset, which is itself a product of our automated pipeline. Our central contribution, as recognized by the reviewer, **is the establishment of the end-to-end World2Minecraft paradigm**, which is a novel framework for translating real-world images into interactive embodied AI environments, **and the scalable, automated process for generating 3D semantic occupancy data**. The dataset's diversity, including its varied lighting, simply serves as one testament to the capability and comprehensiveness of this underlying pipeline. It is a result and a demonstration of our core method's effectiveness, not a primary claim of innovation in itself.
> > >
> > > ## **2. Regarding the newer, better experimental results**
> > > We sincerely thank the reviewer for raising this point. As the **first end-to-end pipeline that translates real-world images into editable virtual scenes for embodied agent training, no directly comparable method existed at the time of our initial submission**. We focused our evaluation on demonstrating: (1) the reconstruction fidelity of World2Minecraft, (2)conduct VLN in scene constructed by World2Minecraft, (3) the quality and scale of the MinecraftOcc dataset, and (4) its utility as both a benchmark and complementary data source. We believed these aspects most directly validated the novelty and effectiveness of our proposed method.
> > >
> > > Based on the insightful feedback from you and other reviewers, we conducted new experiments to compare our "real-to-virtual" reconstruction step against **several recent indoor layout generation methods**, which represent the most relevant solutions **(as they cannot be directly used for embodied AI training and testing)**. We are encouraged that the results, which demonstrate our advantage in this key reconstruction phase, provide strong independent validation of our approach's effectiveness. We will integrate this comparative analysis into the revised manuscript to offer a more comprehensive evaluation.
> > >
> > >
> > > We welcome any further questions and are enthusiastic about the opportunity to discuss our work in greater detail. Wish you every success in your future research!

---

> ### Author Response · Authors · 2025-11-23
>
> ***Part 5***
> ## **4. Response to Weakness 3: Introduction Logic**
> We thank the reviewer for this feedback. We agree that the logic connecting the need for editable simulators to our choice of 3D semantic occupancy can be made more explicit. The intended chain of argument is:
>
> - **Need for Editable Simulators:** Embodied AI research requires high-fidelity, editable simulation environments (unlike static, pre-built ones in Habitat/AI2-THOR).
> - **Limitation of Current Real-to-Sim:** Modern reconstruction techniques (NeRF, 3DGS) create beautiful but non-editable and non-physical neural representations, making them unsuitable for tasks requiring interaction and semantic reasoning.
> - **Our Bridging Solution:** Therefore, we need a representation that is both derived from real-world data and is explicitly structured, editable, and physical. 3D semantic occupancy is this "sweet spot"—it provides dense geometry and semantics that can be directly translated into a voxel-based, programmable environment like Minecraft.
>
> We will revise the introduction to clearly articulate this stepwise logic and justify our core design choice.

---

> > ### Comment · Reviewer_Nuc5 · 2025-11-27
> >
> > Thank you to the authors for the explanation. With this clarification, the logical structure of the introduction is much clearer.

---

### Official Review · Reviewer_7kuq · 2025-10-30

**Soundness:** 3
**Presentation:** 3
**Contribution:** 3
**Rating:** 6
**Confidence:** 4

**Summary:**

This paper introduces World2Minecraft, a pipeline to convert real-world scenes into editable Minecraft environments for embodied AI research using 3D semantic occupancy prediction. The authors find this process is limited by the poor quality of current occupancy models. To address this, they present MinecraftOcc, a large-scale, automatically generated dataset from 156 photorealistic, modded Minecraft scenes (100,165 images) to serve as a challenging benchmark and training resource. Experiments show MinecraftOcc exposes the generalization limits of SOTA models and improves their performance on real-world data (NYUv2) when used for joint training. The utility of the end-to-end pipeline is also shown via a new MinecraftVLN dataset created from the reconstructed scenes.

**Strengths:**

1. The World2Minecraft framework is a clever solution for creating editable, high-fidelity simulation environments, bridging the gap between static real-world scans and blocky, low-fidelity simulators.

2. The MinecraftOcc dataset is a significant contribution, providing a large-scale (100k+ images), high-resolution, and perfectly-annotated data source for training and benchmarking 3D semantic occupancy models.

3. The paper effectively demonstrates MinecraftOcc's value. SOTA models perform poorly on it, highlighting generalization gaps , while joint training with it improves performance on real-world data (NYUv2).

4. The authors validate the full pipeline by creating a MinecraftVLN dataset from the reconstructed scenes and successfully training SFT/RFT agents for navigation tasks .

**Weaknesses:**

1. The World2Minecraft pipeline is not fully automated; its initial outputs were "suboptimal" and required "meticulous manual refinement" to be usable for the VLN dataset, undermining its scalability claims.

2. The paper has two loosely connected parts: the World2Minecraft (real-to-sim) pipeline and the MinecraftOcc (sim-for-training) dataset. The dataset is not generated from the pipeline, which makes the overall story less cohesive.

3. The "Base" MinecraftVLN dataset derived from the pipeline is small (15 scenes) with "simple instructions". The authors had to add 5 external community-built maps to achieve sufficient complexity.

4. The mapping of 1,452 mod-specific classes to 13 standard classes for cross-dataset experiments is coarse and potentially noisy (e.g., "curtains" and "lights" are mapped to "empty"), which could confound the joint-training results.

**Questions:**

1. Can you quantify the "meticulous manual refinement" required for the 15 scenes? How does this manual bottleneck affect the scalability of the World2Minecraft pipeline?

2. Have you considered closing the loop: using World2Minecraft to reconstruct a real-world scene, and then using that reconstructed scene as input for your MinecraftOcc automated data generation pipeline?

3. How do you ensure the coarse 1,452-to-13 class mapping (e.g., "curtains" to "empty") doesn't introduce significant label noise that negatively impacts model training and evaluation?

4. How do you disentangle the models' poor performance on MinecraftOcc (Table 4) between a true "generalization failure" and a simple "domain gap" (i.e., real-world models performing badly on Minecraft's specific visual style)?

---

> ### Author Response · Authors · 2025-11-23
> **Response to Reviewer 7kuq**
>
> ***Part 1***
>
> We thank the reviewer for their positive assessment of our contributions and their constructive questions. Please find our point-by-point responses below.
> ## **1. Response to Weakness 1 and Question 1: "Manual Refinement" & "Automated PipeLine"**
> ***1. About Manual Refinement***
> We appreciate the reviewer's valid concern regarding manual refinements. The required refinements primarily involved **removing floating artifact voxels ("noise blocks")** and **filling minor unpredicted holes** in surfaces—imperfections stemming from limitations in the occupancy prediction algorithm rather than our pipeline itself. To quantitatively address scalability concerns, we conducted a controlled experiment comparing our **"World2Minecraft + Refinement"** workflow against building a comparable scene from scratch. The results, detailed in the below table, demonstrate the efficiency of our approach:
> | Metric | World2Minecraft + Refine | Build from Scratch |
> | :--- | :---: | :---: |
> | **Time (s)** | **70.38** (5.88 + 64.50) | 482.00 |
> | **Total Operations** | **24.50** | 340.00 |
> | Completion Actions | 9.70 | - |
> | Deletion Actions | 7.60 | - |
> | Addition Actions | - | 319.30 |
> | Orientation Adjustments | **7.20** | 20.70 |
>
> Our experiments were conducted on the Base part of the MinecraftVLN, which represents nearly a **7x reduction(70.38s vs. 482.00s)** in required human effort. The refinement process itself was efficient, involving an average of **24.5 operations (compared to 340 for Build from Scratch)** per scene (e.g., ~9.7 hole fillings, ~7.6 noise deletions, ~7.2 orientation adjustments).
>
> ***2. Automated PipeLine***
> Actually, our pipeline is fully automated by design, with manual fine-tuning incorporated specifically to enhance embodied task suitability. As shown in the experiments, our automated pipeline provides a substantial foundation, and the refinement step is a light-weight, post-processing correction that does not undermine its practical utility.

---

> ### Author Response · Authors · 2025-11-23
>
> ***Part 2***
> ## **2. Response to Weakness 2 and Question 2: The Disconnect Between World2Minecraft and MinecraftOcc**
>
> ***1. Why The Disconnect Between World2Minecraft and MinecraftOcc***
> Our development of the World2Minecraft pipeline revealed a **critical dependency on the quality of 3D reconstruction**. We observed that the performance of reconstruction tasks, and 3D vision tasks in general, is **heavily constrained** by their training data, which is predominantly limited to  **ScanNet dataset**. The prohibitive cost of creating large-scale, high-quality 3D datasets presents a major bottleneck for the field.
>
> This insight directly motivated the creation of the MinecraftOcc dataset. We recognized that Minecraft, enhanced with photorealistic shaders and mods, offers a unique and cost-effective platform to simulate highly realistic environments. By leveraging this, we can procedurally generate a massive-scale dataset for 3D semantic occupancy prediction. Therefore, creating MinecraftOcc is a **direct and strategic response** to the core limitation we identified while building World2Minecraft, aimed at propelling the entire field of 3D scene understanding forward.
>
> ***2. The Closing Loop***
> Thanks for your advice. This is a compelling idea for future work. The primary reason we did not implement this closed loop is that the current reconstruction quality is suboptimal. Using these imperfect reconstructions as a source for generating MinecraftOcc would propagate errors and result in a lower-quality, noisy dataset. Our current approach of using high-quality, human-built scenes ensures MinecraftOcc serves as a clean and challenging benchmark. Improving reconstruction to enable this loop is a key direction for future research.

---

> ### Author Response · Authors · 2025-11-23
>
> ***Part 3***
> ## **3. Response to Weakness 3: Small and Simple Base Part of MinecraftVLN**
> As correctly noted by the reviewer, the base part of MinecraftVLN indeed consists of relatively small scenes with simple instructions. This is because, for practical considerations, we constructed this base portion using the **OccScanNet dataset**, where scenes are typically confined to **individual small rooms**. To address this limitation and enhance the dataset's diversity, we introduced the external portion. This expansion precisely underscores the critical need for larger and more varied 3D scenes in the field.
>
> Furthermore, we would like to emphasize that the primary objective of creating the MinecraftVLN dataset was to **validate the feasibility of conducting VLN tasks** within environments reconstructed by our World2Minecraft pipeline. This capability has been successfully demonstrated not only in the base portion but also, more significantly, within the larger and more complex extend scenes.
>
> ## **4. Response to Weakness 4 and Question 3: Coarse Class Mapping**
> Thank you for this careful observation. The mapping of 1,452 mod-specific classes to 13 standard classes was performed to ensure a **fair and direct comparison with existing dataset and methods**, which are all trained and evaluated on this standard label set (NYUv2, OccScanNet). This is a standard practice in cross-dataset evaluation.
>
> Upon careful examination of the NYUv2 dataset, we observed that it inherently **overlooks numerous fine-grained classes** and does **not assign semantic labels to them**. An example is objects like curtains, which are not distinctly categorized. Following this, we have consistently mapped such classes to the "empty" label to maintain dataset consistency and enable fair comparisons, which affects all methods equally in our comparative benchmarks. Our contribution **extends beyond the MinecraftOcc** dataset itself to include a **automated pipeline** for constructing 3D semantic occupancy prediction datasets, which enables seamless generation of fine-grained occupancy datasets supporting up to 1,453 distinct categories.

---

> ### Author Response · Authors · 2025-11-23
>
> ***Part 4***
> ## **5. Response to Question 4: "generalization failure" and "domain gap"**
>
> We appreciate the reviewer's insightful question regarding the root cause of the performance drop on MinecraftOcc. We argue this stems from a fundamental generalization failure in 3D structural understanding, not merely a superficial visual domain gap, for two key reasons:
>
> - **Failure in Geometric Reasoning:** The primary challenge in MinecraftOcc is its inherent discrete, voxelized geometry—a fundamentally different 3D representation from the continuous surfaces common in real-world scans (NYUv2, ScanNet). SOTA models, **heavily trained on these specific datasets**, have **overfitted** to their particular geometric priors. The drastic performance drop occurs because these models fail to transfer their learned knowledge to a different yet perfectly valid 3D structural domain. This indicates a brittle understanding of 3D space itself.
>
> - **Underlying Data Scarcity**: The core issue enabling this overfitting is the severe scarcity of large-scale, diverse 3D occupancy datasets. With models seeing limited structural variations (mostly from NYUv2/ScanNet), they cannot learn robust, domain-agnostic 3D representations. Our claim is **supported** by the subsequent experiment (Table 5 in manuscript): **jointly training models with MinecraftOcc improves their performance** on the real-world NYUv2 benchmark. If the issue were purely a visual domain gap, incorporating stylized Minecraft data would not enhance performance on real images. This confirms that MinecraftOcc provides complementary 3D structural knowledge that improves fundamental geometric reasoning, underscoring its value in addressing the field's data bottleneck.

---

### Official Review · Reviewer_HVMZ · 2025-10-31

**Soundness:** 2
**Presentation:** 2
**Contribution:** 2
**Rating:** 4
**Confidence:** 2

**Summary:**

The paper introduces a real-to-simulation framework, World2Minecraft, which converts real-world observations into a Minecraft environment through semantic occupancy prediction. The authors further utilize the constructed Minecraft scenes to generate samples for vision-language navigation and additional data for training semantic occupancy prediction models. Experimental results show that this augmentation strategy holds potential to enhance task performance.

**Strengths:**

* Leveraging simulation to augment real-world task performance through real-to-sim techniques is a promising direction.
* Developing an automated real-to-sim pipeline is highly valuable.

**Weaknesses:**

* Unclear real-to-sim pipeline： The process of converting real scenes into Minecraft is insufficiently detailed. It is unclear how furnitures are selected, how their poses are estimated, how occlusions are handled, and how the approach compares to existing real-to-sim methods. This raises doubts about how automated the proposed pipeline truly is.
* If the pipeline relies on an existing furniture set (e.g., MOD assets), the method becomes similar to prior real-to-sim approaches that depend on existing object set. It is not obvious why constructing the environment specifically in Minecraft provides clear advantages. Additionally, the physics in Minecraft appears irrelevant for the presented tasks.
* Manual refinements reduce practicality: For the VLN task, the authors mention manually refining 15 environments due to data quality issues, but do not detail the required operations or effort. This raises concerns about scalability. Furthermore, if data quality is limited, it is unclear whether using all constructed scenes benefits the semantic occupancy task—does every scene require manual refinement as well?

**Questions:**

* Task-specific scene usage: Why are only 15 manually refined scenes used for the VLN task, while all constructed scenes are utilized for semantic occupancy prediction? What specific refinements are required for VLN, and why are they unnecessary for occupancy prediction?
* Scope of reconstruction: Since furniture assets come from an existing asset set, does the proposed pipeline only reconstruct the structural layout of the environment? If so, what are the key advantages over prior real-to-sim frameworks that similarly rely on existing furniture databases?

---

> ### Author Response · Authors · 2025-11-23
> **Response to Reviewer HVMZ**
>
> ***Part 1***
>
> We thank the reviewer for the constructive feedback and for recognizing the value of our automated real-to-sim pipeline. We appreciate the opportunity to clarify the technical details of our reconstruction process, the rationale behind using Minecraft, and the specifics of our experimental setup. Below, we address the concerns point-by-point.
>
> ## **1. Response to Weakness 1 and Question 2: "Unclear Real-to-Sim pipeline" and "Only Need Layout"**
> We apologize for the lack of detail regarding the specific operations in the reconstruction pipeline. We have now revised the manuscript to include a detailed algorithm description. Here is the clarification regarding your specific questions:
>
> ***1. Unclear Real-to-Sim pipeline***
>
> **1) Furniture Selection & Pose Estimation.** We do not simply place objects based on semantic labels. Instead, we utilize a shape-aware template matching approach:
> - **Instance Isolation:** We isolate the semantic occupancy grid for individual objects from the global prediction.
> - **Selection:** We voxelize the OBJ models from our furniture library and calculate the structural similarity (IoU) between the predicted object occupancy and the library assets to select the best match.
> - **Pose Estimation:** We rotate the candidate furniture asset to various orientations and compare it against the predicted occupancy. The orientation yielding the highest overlap is selected as the final pose.
>
> **2) Handling Occlusion.** This is a core advantage of using Semantic Occupancy Prediction as the intermediate representation. The occupancy network is explicitly trained to complete geometries and predict occluded voxels based on learned priors. This inherent capability to reconstruct complete structures from partial observations explains why occupancy representation has been widely adopted in autonomous driving, where reliably inferring occluded areas is critical for safety. By leveraging this strength, our system handles occlusion at the prediction stage before asset matching begins, producing more complete and robust scene reconstructions.
>
> ***2. Comparison with Existing Methods***
> Unlike prior works that primarily rely on layout estimation or bounding boxes, our method provides a more geometrically accurate reconstruction by directly matching the predicted semantic occupancy against 3D furniture models. This occupancy-based template matching enables superior scale consistency and shape alignment, as validated in the following experiment. As shown in the results table, our approach achieves **the highest scores in Semantic Integrity (0.913) and Visual Realism (6.145)**, demonstrating its ability to preserve fine-grained geometric details from RGB images to the final scene—an end-to-end capability not achieved by existing methods.
> | Method | OOB Rate $\downarrow$ | Collision Count $\downarrow$ | Semantic Integrity $\uparrow$ | Visual Realism $\uparrow$ | Scene Completeness $\uparrow$ | Aesthetic Atmosphere $\uparrow$ |
> | :--- | :---: | :---: | :---: | :---: | :---: | :---: |
> | LayoutGPT | 0.279 | 4.5 | 0.689 | 5.000 | 3.856 | 4.582 |
> | I-Design | 0.423 | **0** | 0.884 | 6.001 | 4.734 | 5.352 |
> | LayoutVLM | **0** | 0.9 | 0.348 | 3.625 | 2.270 | 2.708 |
> | **World2Minecraft** | 0.024 | 0.2 | **0.913** | **6.145** | **5.186** | **6.022** |
>
> ***3. Why not Only Layout?***
> Our pipeline goes beyond structural layout. Relying solely on layout would lead to two fundamental problems that our occupancy-based approach directly solves:
> * **Shape & Scale Awareness:** Our pipeline fundamentally requires semantic occupancy, as structural layout alone is insufficient for realistic reconstruction. While a layout defines room boundaries, it fails to capture the critical scale and shape of individual objects. For instance, it cannot distinguish a single bed from a king-size bed—a vital difference that directly impacts navigability, as misplacing an oversized bed could block a virtual agent's path. Semantic occupancy provides the precise 3D geometry of each instance, enabling our shape-aware template matching to select library assets that accurately reflect real-world scale and form. This ensures the reconstructed environment is not only visually faithful but also functionally plausible for downstream tasks.
> * **Instance-Level Matching:** Because we reconstruct the **occupancy** first, we obtain the specific voxel shape of each instance. This allows our template matching to select assets that match the *geometry* of the real-world object, not just its semantic class. This provides a denser, more informative reconstruction than layout-only approaches.

---

> ### Author Response · Authors · 2025-11-23
>
> ***Part 2***
>
> ## **2. Response to Weakness 2: "Why Minecraft?" & Physics Relevance**
>
> We chose Minecraft for two primary reasons that offer distinct advantages over other simulators:
>
> - **Visual and Functional Fidelity via Mods:** Minecraft serves as a highly extensible platform. Beyond employing high-resolution texture packs and shaders for enhanced visual realism, its true strength lies in the ability to load mods that fundamentally redefine the world's physics and interactive logic. This capability to arbitrarily customize environmental rules, which is not easily replicated in conventional simulators, is particularly valuable for creating diverse and complex training environments for downstream tasks.
> - **Physics for Downstream Tasks:** Minecraft offers a robust and efficient physics engine that simulates gravity, collision detection, agent walking mechanics, and field-of-view rendering out-of-the-box. As demonstrated in our experiments on the **VLN** task, this capability is crucial. It allows us to directly deploy agents in a reconstructed environment where navigation constraints (walls, obstacles, gravity) are automatically enforced, facilitating the training of agents that can navigate complex real-world topologies.

---

> ### Author Response · Authors · 2025-11-23
>
> ***Part 3***
>
> ## **3. Response to Weakness 3 and Question 1: "Manual Refinements"**
>
>
> ***1. Why are only 15 refined scenes used for VLN, while all scenes are used for Semantic Occupancy?***
>
> There is a fundamental difference in how the data is used for these two tasks:
>
> * **MinecraftVLN (Requires Refinement):** This dataset is designed to **verify if agents can navigate in *reconstructed real-world* environments**. The output of any reconstruction algorithm inevitably contains noise (e.g., floating voxels or artifacts). In a navigation task, a single **noise block can block a doorway or trap an agent**, making the environment unplayable. Therefore, we performed manual refinement (removing noise blocks, filling holes) to ensure navigability.
> * **MinecraftOcc (No Refinement Needed):** This dataset aims to *train* the Semantic Occupancy prediction model. For this, we utilize high-quality **community-built Minecraft scenes**. These scenes are constructed by humans from scratch and are **inherently "clean" (no reconstruction noise or holes)**. We generate training pairs (images and ground truth occupancy) from these clean scenes. Since we are not reconstructing these scenes from the real world, but rather using them as a source of synthetic training data, no manual refinement is required.
>
> ***2. What specific refinements were required?***
> The manual refinements were light-weight and involved only three types of operations:
> - **Deletion**: Removing floating artifact voxels ("noise blocks").
> - **Completion**: Filling minor unpredicted holes in surfaces.
> - **Adjustment**: Correcting a small number of object orientations.
>
> ***3. How much effort was required, and is it scalable?***
> The required effort was minimal. To quantitatively address the concern, we conducted a controlled experiment comparing our **"World2Minecraft + Refinement"** workflow against building a comparable scene from scratch. The results, detailed in the below table, demonstrate the efficiency of our approach:
> | Metric | World2Minecraft + Refine | Build from Scratch |
> | :--- | :---: | :---: |
> | **Time (s)** | **70.38** (5.88 + 64.50) | 482.00 |
> | **Total Operations** | **24.50** | 340.00 |
> | Completion Actions | 9.70 | - |
> | Deletion Actions | 7.60 | - |
> | Addition Actions | - | 319.30 |
> | Orientation Adjustments | **7.20** | 20.70 |
>
> Our experiments were conducted on the Base part of the MinecraftVLN, which represents nearly a **7x reduction(70.38s vs. 482.00s)** in required human effort. The refinement process itself was efficient, involving an average of **24.5 operations (compared to 340 for Build from Scratch)** per scene (e.g., ~9.7 hole fillings, ~7.6 noise deletions, ~7.2 orientation adjustments).
>
> ***4. Does every scene require manual refinement?***
> Manual refinement was exclusively performed on the 15 **MinecraftVLN** scenes **to ensure robust navigability for VLN tasks**, where even minor reconstruction artifacts like floating blocks could obstruct paths or holes could trap agents. In contrast, the **MinecraftOcc** dataset **required no manual refinements**, as it was automatically generated from pristine, human-designed community environments. This distinction highlights that refinement was a task-specific necessity for VLN evaluation, not a general requirement of our data generation pipeline.

---

### Author Response · Authors · 2025-12-03
**Summary of Rebuttal and Response to Reviewers**

Dear Area Chair,

We thank the reviewers for their time and constructive feedback. During the rebuttal period, we have actively engaged with the reviewers, provided detailed clarifications, and added new comparative experiments. We would like to provide a brief summary of our responses and the current status.
## **1. The Contribution and Strength**
- **First End-to-End Pipeline (World2Minecraft & MinecraftVLN):** To the best of our knowledge, we are **the first to establish a complete pipeline that spans the entire lifecycle**: from several real-world images to scene reconstruction **(World2Minecraft)**, virtual environment instantiation, and subsequent agent training/testing. We validate the practical utility of this pipeline by constructing the **MinecraftVLN** dataset to conduct Vision-and-Language Navigation tasks within the reconstructed scenes.
- **Scalable Data Generation & Large-scale Benchmark (MinecraftOcc):** We propose a novel **automated framework for 3D semantic occupancy data generation** and present **MinecraftOcc**, a large-scale dataset comprising **over 100,000 images**. This benchmark not only serves as effective training data for enhancing robustness but also exposes the generalization limits of existing SOTA methods.
## **2. The Reviewers Status Update**

* **Reviewer Nuc5 (Score raised: 2 $\to$ 6):** We are grateful that Reviewer Nuc5 has acknowledged our clarifications regarding the **innovation of our pipeline** and **the value of the MinecraftOcc dataset**. Following our detailed explanation of the technical challenges and new experimental comparisons, the reviewer was convinced and **raised their score to 6, which is  before the information leak**.

* **Reviewers HVMZ, X8ju, 7kuq (Pending Final Feedback):** We have comprehensively addressed all concerns raised by these reviewers in our official responses and through new experiments. Although they **have not yet responded**, we are confident that our revisions effectively resolve their issues:
    * **Reviewer 7kuq (Score: 6):** We have **clarified the misunderstandings** arising from the text.
    * **Reviewer X8ju (Score: 4):** The reviewer explicitly **stated a willingness to increase the score** if their concerns were clarified. We are confident that our new experimental results and explanations have fully met their criteria.
    * **Reviewer HVMZ (Score: 4):** We have conducted additional experiments to substantiate our claims and address their specific questions regarding the pipeline details.
## **3. Response to Key Concerns**
We have addressed the key concerns as follows:
- **Justification of Semantic Occupancy vs. Layout (Addressed HVMZ & Nuc5 & X8ju):**
  - Reviewers asked why we utilize occupancy rather than simple **layout estimation**. We clarified that layout alone **fails to capture object scale, shape, and instance-level geometry** (e.g., distinguishing a king-size bed from a single bed), which is critical for navigability.
  - **Evidence.** We provided a new table showing our method significantly outperforms layout-based baselines (e.g., LayoutGPT, LayoutVLM) in Semantic Integrity **(0.913 vs 0.689)** and Visual Realism **(6.145 vs 5.000)**.
- **Pipeline Details & Scalability (Addressed HVMZ & Nuc5 & 7kuq):**
  - Concerns were raised regarding the **"manual refinement"** required. We clarified that refinement is only needed for the VLN task (to remove navigation-blocking artifacts) and is not a bottleneck for the general pipeline.
  - **Evidence.** We quantified this effort: it takes only **~70s** per scene using our tool versus **482s** to build from scratch **(a ~7x reduction)**, proving the pipeline is highly automated and scalable. We also added missing algorithmic details regarding furniture selection and pose estimation.
- **Dataset Value & Automated Pipeline (Addressed Nuc5 & X8ju):**
We addressed the concern regarding the improvement on NYUv2 by clarifying that our contribution extends beyond the dataset itself:
  - **Automated Data Generation Pipeline:** We propose a novel, scalable **framework for collecting 3D semantic occupancy data**, directly addressing the critical data scarcity issue in the field.
  - **Dual Role of MinecraftOcc**: It serves as both effective **supplementary training data** (providing complementary structural knowledge to improve real-world performance) and a **challenging benchmark** (exposing the generalization limits of SOTA models).

We trust that these clarifications and new results demonstrate the solidity and contribution of our work, allowing for a fair and reasonable assessment.

Best regards,

The Authors

---

### Meta-Review · Area_Chair_qTd2 · 2026-01-05

**Summary:**

This paper presents World2Minecraft, a pipeline for converting real world scenes to minecraft worlds. The resulting minecraft worlds enjoy the benefits of any minecraft environment -- high quality visuals with an editable/interactive scene. The useful of the pipeline is validated with two applications. The first is MinecraftVLN, where 15 scenes from the EmbodiedOcc-ScanNet dataset and 5 community-created Minecraft scenes are used to construct a Vision-Language Navigation dataset. The second is MinecraftOcc, an occupancy prediction dataset. Both experiments demonstrate the validity of the pipeline -- agents are able to navigated within MinecraftVLN and MinecraftOcc provides data the transfers to real-world datasets.

While reviewers brought up valid concerns in the initial reviewer, they also saw the potential value of the paper. It is the opinion of the AC that the concerns have been adequately addressed.

**Reviewer Concerns:**

## Reviewer HVMZ

- Clarity of the real-to-sim pipeline. I believe this is addressed.
- Why use Minecraft? I believe this is addressed.
- Requirement of manual refinement. I believe this is partially addressed.

## Reviewer 7kuq

- The pipeline is not fully automated (i.e. it requires manual refinement). I believe this is partially addressed.
- Lack of clarity around the two parts of the paper -- World2Minecraft vs MinecraftOcc.  I in general understand the motivation for this dataset, but, as far as I can tell, the authors don't use MinecraftOcc in service of creating World2Minecraft -- doing so would result in a clearer story. Thus I believe this is partially addressed.
- Scale of the MinecraftVLN dataset. I believe this is addressed.
- The 1,452-to-13 class mapping is likely noisy. I believe this is addressed.

## Reviewer Nuc5

- Lack of innovation in the method. I believe this is addressed.
- Value and diversity of the dataset. I believe this is addressed.
- Clarity of motivation in the introduction. I believe this is addressed.
- Strength of results. I believe this is addressed.

## Reviewer X8ju

- Is occupancy needed for reconstruction? Why not just layout? I believe this is resolved.
- Training on MinecraftOcc only provides a small increase in performance. I believe this is resolved.
- What is the value of MinecraftVLN? I believe this is resolved.

**Reviewer Scores:**

Reviewer HVMZ -- 6

Reviewer 7kuq -- 6

Reviewer Nuc5 -- 6

Reviewer X8ju -- 6

---

### Decision · Program_Chairs · 2026-01-26

Accept (Poster)